# Compact Memory for Continual Logistic Regression

**Yohan Jung**[1]    **Hyungi Lee**[2]*    **Wenlong Chen**[3]*    **Thomas Möllenhoff**[1]
**Yingzhen Li**[3]    **Juho Lee**[4]    **Mohammad Emtiyaz Khan**[1]

[1]RIKEN Center for AI Project    [2]Kookmin University
[3]Imperial College London    [4]KAIST
*Work performed in part during internship at the RIKEN Center for AI Project.

## Abstract

Despite recent progress, continual learning still does not match the performance of batch training. To avoid catastrophic forgetting, we need to build compact memory of essential past knowledge, but no clear solution has yet emerged, even for shallow neural networks with just one or two layers. In this paper, we present a new method to build compact memory for logistic regression. Our method is based on a result by Khan and Swaroop [2021] who show the existence of optimal memory for such models. We formulate the search for the optimal memory as Hessian-matching and propose a probabilistic PCA method to estimate them. Our approach can drastically improve accuracy compared to Experience Replay. For instance, on Split-ImageNet, we get $60\%$ accuracy compared to $30\%$ obtained by replay with memory-size equivalent to $0.3\%$ of the data size. Increasing the memory size to $2\%$ further boosts the accuracy to $74\%$, closing the gap to the batch accuracy of $77.6\%$ on this task. Our work opens a new direction for building compact memory that can also be useful in the future for continual deep learning.

## 1 Introduction

Continual learning aims for a continual lifelong acquisition of knowledge which is challenging because it requires a delicate balance between the old and new knowledge [Mermillod et al., 2013]. New knowledge can interfere with old knowledge [Sutton, 1986] and lead to catastrophic forgetting [Kirkpatrick et al., 2017]. Thus, it is necessary to compactly memorize all the past knowledge required in the future to quickly learn new things. This is a difficult problem and no satisfactory solution has been found yet, even for simple cases such as logistic regression and shallow neural networks. The problem is important because solving it can also drastically reduce the cost and environmental impact of batch training which requires access to all of the data all the time.

The most straightforward method to build memory is to simply store old models and data examples, but such approaches do not perform as well as batch training. For instance, weight-regularization uses old parameters to regularize future training [Kirkpatrick et al., 2017, Li and Hoiem, 2017, Lee et al., 2017, Zenke et al., 2017, Ebrahimi et al., 2020, Ritter et al., 2018, Nguyen et al., 2018]. While this is memory efficient, it often performs worse than replay method that simply store old raw data for reuse in the future [Rolnick et al., 2019, Lopez-Paz and Ranzato, 2017, Chaudhry et al., 2019, Titsias et al., 2020, Buzzega et al., 2020, Pan et al., 2020, Scannell et al., 2024]. Many replay-based methods have been devised to effectively select past data points based on gradient [Aljundi et al., 2019a, Yoon et al., 2022], interference [Aljundi et al., 2019b], information-theoretic metric [Sun et al., 2022], and Bilevel optimization [Borsos et al., 2020]. The memory cost of replay can be reduced by combining it with weight regularization, but so far the performance still lags behind the batch training [Daxberger et al., 2023]. The goal of this work is to propose new methods to build compact memory for the

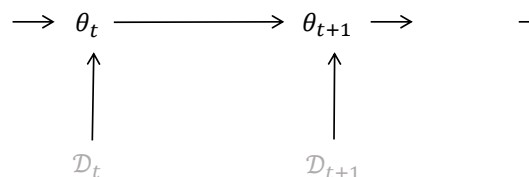
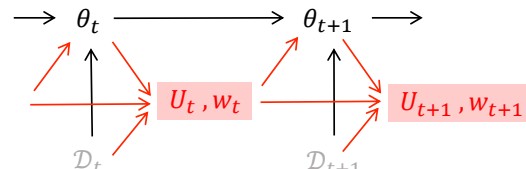

Figure 1: Standard continual-learning methods, such as those using weight regularization, use past parameters $\boldsymbol{\theta}_t$ to update $\boldsymbol{\theta}_{t+1}$ for the new task $\mathcal{D}_{t+1}$ (shown on the left). We propose a new method (shown on the right) that also builds compact memory and reuses it to continually learn. The memory consists of a set $\mathbf{U}_t$ of memory vectors and a weight vector $\mathbf{w}_t$. These are used to update $\boldsymbol{\theta}_{t+1}$ when new $\mathcal{D}_{t+1}$ arrives. Afterward, the memory is updated to get the new $(\mathbf{U}_{t+1}, \mathbf{w}_{t+1})$.

simplest non-trivial case of logistic regression. With this, we hope to gain new insights to be able to do the same in the future for continual deep learning.

We build upon the work of Khan and Swaroop [2021] who show the existence of compact memory for logistic regression but do not give a practical method to estimate them. They propose a prior called the Knowledge-adaptation prior (K-prior) which uses a 'memory' set to approximate the full-batch gradient. More precisely, given parameter $\boldsymbol{\theta}_t$ after training until task $t$, a subset $\mathcal{M}_t$ of the past data $\mathcal{D}_{1:t} = \{\mathcal{D}_1, \mathcal{D}_2, \ldots, \mathcal{D}_t\}$ is used to construct the K-prior $\mathcal{K}_t(\boldsymbol{\theta}; \mathcal{M}_t)$ such that the following holds:

$$\sum_{j=1}^{t} \nabla \bar{\ell}_j(\boldsymbol{\theta}; \mathcal{D}_j) \approx \nabla \mathcal{K}_t(\boldsymbol{\theta}; \mathcal{M}_t), \tag{1}$$

where $\bar{\ell}_j(\boldsymbol{\theta}; \mathcal{D}_j)$ denotes the loss over examples in $\mathcal{D}_j$. Khan and Swaroop [2021, App. A] show that, for logistic regression, it is also possible to construct an *optimal* memory by using the singular vectors of the feature matrix. The optimality refers to the fact that the memory can, in theory, *perfectly* recover full-batch gradient, that is, the approximation in Eq. 1 becomes an equality. The optimal memory size depends on the rank of the feature matrix, not on the size of whole $\mathcal{D}_{1:t}$, yielding a compact optimal memory. The main issue with the method is that it requires access to the whole $\mathcal{D}_{1:t}$ to construct the memory which is not possible when learning continually. Our goal here is to propose a practical alternative to estimate such memory for continual logistic regression.

In this paper, we reformulate the search for the optimal memory as Hessian-matching and propose a practical method to estimate them (Fig. 1). We are motivated by the fact that, in linear regression, estimation of optimal memory can be reformulated as Hessian matching. We generalize this to logistic regression by posing it as a maximum likelihood estimation problem which uses the classical Probabilistic Principal Component Analysis [PPCA; Tipping and Bishop, 1999a]. The method does not yield perfect gradient reconstruction anymore, but can still drastically improve performance compared to replay of raw old data. For instance, on the Split-ImageNet dataset, we obtain 60% accuracy compared to the 30% accuracy obtained by replay with memory-size equivalent to 0.3% of the data size. A small memory size of 2% of the data size is sufficient to obtain an accuracy of 74%, closing the gap to the batch accuracy of 77.6% on this task. Our work opens a new direction for building compact memory that can also be useful in the future for continual deep learning. Our code is available at https://github.com/team-approx-bayes/compact_memory_code.

## 2 Compact Memory for Continual Learning

We consider continual learning for supervised problems where the data of the $j$'th task consists of $N_j$ input-output pairs of form $(\mathbf{x}_j, y_j)$. The goal is to learn the parameter vector $\boldsymbol{\theta}$ whose predictions closely match the true labels $y_j$. The empirical loss over task $j$ is defined as

$$\bar{\ell}_j(\boldsymbol{\theta}; \mathcal{D}_j) = \sum_{i \in \mathcal{D}_j} \mathcal{L}\left(y_j, \hat{y}(f_{\boldsymbol{\theta}}(\mathbf{x}_j))\right), \tag{2}$$

where $\mathcal{L}(y, \hat{y})$ is chosen to be the cross-entropy loss over predictions $\hat{y}$ obtained by passing the logits $f_{\boldsymbol{\theta}}(\mathbf{x})$ through the softmax function. As discussed earlier, learning continually requires a delicate

balance between old and new knowledge. Tasks may arrive one at a time and there may be little flexibility to revisit them again. This is unlike the standard 'batch' training where the goal is to obtain the parameter $\boldsymbol{\theta}_T^*$ obtained by jointly training over the sum $\sum_{j=1}^T \bar{\ell}_j$ over all $T$ tasks. Such batch training typically requires access to all the data all the time. Continual learning eliminates this requirements and aim to recover $\boldsymbol{\theta}_T^*$ by incrementally training of tasks one at a time.

Continual learning however is extremely challenging, even for simple cases such as logistic regression and shallow neural networks. If new task interferes with the old knowledge, there is a chance the model may catastrophically forget the previously acquired knowledge. Because the tasks are not revisited, the model may never be able to relearn it. It is therefore important to compactly memorize all essential knowledge that may be needed to learn new things in the future. This is challenging because the future is uncertain and new tasks may be very different from old ones. No satisfactory solution has been found yet to build compact memory that facilitates accurate continual learning.

Currently, there are two most popular methods to build memory and both do not perform as well as batch training. The first method is to store the old model parameters $\boldsymbol{\theta}_t$ after training until task $t$ and reuse it to train over the next task. For example, we can use a quadratic weight-regularizer:

$$\boldsymbol{\theta}_{t+1} \leftarrow \arg\min_{\boldsymbol{\theta}} \bar{\ell}_{t+1}(\boldsymbol{\theta}; \mathcal{D}_{t+1}) + \tfrac{1}{2}\delta_t\|\boldsymbol{\theta} - \boldsymbol{\theta}_t\|^2, \tag{3}$$

where $\delta_t > 0$ is the regularization parameter. This is memory efficient because the memory cost does not grow with tasks and requires storing only one model parameters. It is also possible to use multiple checkpoints as long as the memory does not grow too fast.

The second method is to store old raw data, which is often not the most memory efficient approach. This is because the memory size could grow larger as new tasks are seen. However, in practice, the method often performs much better than weight regularization. The method is also easy to implement within standard stochastic training pipeline. For instance, given a set $\mathcal{E}_t$ of replay examples, we can simply us the following update,

$$\boldsymbol{\theta}_{t+1} \leftarrow \arg\min_{\boldsymbol{\theta}} \bar{\ell}_{t+1}(\boldsymbol{\theta}; \mathcal{D}_{t+1}) + \sum_{j \in \mathcal{E}_t} \mathcal{L}\left(y_j, \hat{y}(f_{\boldsymbol{\theta}}(\mathbf{x}_j))\right). \tag{4}$$

Another alternative is to use a *function-space* regularizer, defined as $\mathcal{L}(\hat{y}(f_{\boldsymbol{\theta}_t}(\mathbf{x}_j)), \hat{y}(f_{\boldsymbol{\theta}}(\mathbf{x}_j)))$ where instead of predicting the true labels $y_j$, we predict prediction $\hat{y}(f_{\boldsymbol{\theta}_t}(\mathbf{x}_j))$ of $\boldsymbol{\theta}_t$. This also boosts performance in practice [Titsias et al., 2020, Buzzega et al., 2020, Pan et al., 2020, Scannell et al., 2024]. The memory efficiency of all of these methods can be reduced by combining them with weight regularization, which brings us to the Knowledge-adaptation prior of Khan and Swaroop [2021].

As discussed earlier, K-prior uses both $\boldsymbol{\theta}_t$ and a memory set $\mathcal{M}_t$ to approximate the full-batch gradient over the past tasks. The memory set consists of $K_t$ vectors, that is, $\mathcal{M}_t = (\mathbf{u}_{1|t}, \mathbf{u}_{2|t}, \ldots, \mathbf{u}_{K_t|t})$ where $\mathbf{u}_{k|t}$ is the $k$'th memory vector. K-prior uses a combination of the weight-space and function-space regularizer to approximate the full-batch gradient. For instance, consider a regularized linear-regression problem with loss $\mathcal{L}(y_j, \hat{y}(\boldsymbol{\phi}_j^\top \boldsymbol{\theta})) = \frac{1}{2}(y_j - \boldsymbol{\phi}_j^\top \boldsymbol{\theta})^2$ for each $(\boldsymbol{\phi}_j, y_j)$ pair, with a regularizer $\frac{1}{2}\delta_t\|\boldsymbol{\theta}\|^2$. For this model, we can use the following combination:

$$\mathcal{K}_t(\boldsymbol{\theta}; \mathcal{M}_t) = \tfrac{1}{2}\delta_t\|\boldsymbol{\theta} - \boldsymbol{\theta}_t\|^2 + \sum_{k=1}^{K_t} \tfrac{1}{2}\left(\mathbf{u}_{k|t}^\top \boldsymbol{\theta}_t - \mathbf{u}_{k|t}^\top \boldsymbol{\theta}\right)^2. \tag{5}$$

The first term ensures that new $\boldsymbol{\theta}$ are close to old $\boldsymbol{\theta}_t$ while the second term ensures the same for their predictions. We then use the following update for the next task,

$$\boldsymbol{\theta}_{t+1} \leftarrow \arg\min_{\boldsymbol{\theta}} \bar{\ell}_{t+1}(\boldsymbol{\theta}; \mathcal{D}_{t+1}) + \mathcal{K}_t(\boldsymbol{\theta}; \mathcal{M}_t). \tag{6}$$

Khan and Swaroop [2021] prove that, for generalized linear models, such an update can perfectly recover the exact gradient when $\mathcal{M}_t = \mathcal{D}_{1:t}$ and $\boldsymbol{\theta}_t = \boldsymbol{\theta}_t^*$ (the batch solution over $\mathcal{D}_{1:t}$). Choosing a subset of $\mathcal{D}_{1:t}$ as the memory set yields an approximation of the gradient whose accuracy can be controlled by carefully choosing the memory set, as long as $\boldsymbol{\theta}_t$ is close enough to $\boldsymbol{\theta}_t^*$.

Choosing $\mathcal{M}_t \subset \mathcal{D}_{1:t}$ however is not the best choice. Khan and Swaroop [2021, App. A] show that it is also possible to construct a much more compact *optimal* memory set which can recover prefect gradients even when $|\mathcal{M}_t| \ll |\mathcal{D}_{1:t}|$. This can be done by setting $\mathbf{u}_{k|t}$ to be the left singular vectors of the feature matrix. To be precise, denoting by $\boldsymbol{\Phi}_{1:t}$ the feature matrix of containing features of inputs in $\mathcal{D}_{1:t}$ as columns, we compute the following singular-value-decomposition (SVD),

$$\boldsymbol{\Phi}_{1:t} = \mathbf{U}_t^* \mathbf{S}_t^* (\mathbf{V}_t^*)^\top, \text{ where } \mathbf{U}_t^* = (\mathbf{u}_{1|t}^*, \mathbf{u}_{2|t}^*, \ldots \mathbf{u}_{K_t^*|t}^*) \text{ and } \mathbf{V}_t^* = (\mathbf{v}_{1|t}^*, \mathbf{v}_{2|t}^*, \ldots, \mathbf{v}_{K_t^*|t}^*). \tag{7}$$

The left and right singular vectors are denoted by $\mathbf{u}^*_{k|t}$ and $\mathbf{v}^*_{k|t}$ respectively. The rank is denoted by $K^*_t$ and $\mathbf{S}^*_t$ is a diagonal matrix containing all $K^*_t$ non-zero singular values denoted by $s^*_{k|t}$. Khan and Swaroop [2021, App. A] define the optimal memory to be $\mathcal{M}_t = \mathbf{U}^*_t$. Then, they construct the following optimal K-prior, denoted by $\mathcal{K}^*_t(\boldsymbol{\theta}; \mathbf{U}^*_t)$, for binary logistic regression,

$$\mathcal{K}^*_t(\boldsymbol{\theta}; \mathbf{U}^*_t) = \tfrac{1}{2}\delta\|\boldsymbol{\theta} - \boldsymbol{\theta}_t\|^2 + \sum_{k=1}^{K^*_t} w^*_{k|t}\, \mathcal{L}\left(\hat{y}((\mathbf{u}^*_{k|t})^\top \boldsymbol{\theta}_t)\,,\, \hat{y}((\mathbf{u}^*_{k|t})^\top \boldsymbol{\theta})\right), \tag{8}$$

and show that there exist scalars $w^*_{k|t}$ for which it perfectly recovers the full-batch gradient at $\boldsymbol{\theta}$. Here, $\hat{y}(f)$ denotes the prediction function which is the sigmoid function for binary logistic regression, that is, $\hat{y}(f) = \sigma(f) = 1/(1 + \exp(-f))$. The result shows that the optimal memory size is equal to $K^*_t$ and perfect gradient recovery is possible by simply comparing $K^*_t$ predictions at old and new models. This can be extremely cheap compared to the full batch gradient for long data streams whose intrinsic dimensionality is much smaller.

The main issue with the method is that computing $w^*_{k|t}$ for each $\boldsymbol{\theta}$ is infeasible in practice because it requires access to all of the past data. Specifically, the optimal value is given by

$$w^*_{k|t} = \frac{s^*_{k|t}(\mathbf{v}^*_{k|t})^\top \mathbf{d}_x}{\sigma(\mathbf{u}^\top_{k|t}\boldsymbol{\theta}) - \sigma(\mathbf{u}^\top_{k|t}\boldsymbol{\theta}_t)} \tag{9}$$

which require computation of the vector $\mathbf{d}_x$ whose length is equal to the data size $|\mathcal{D}_{1:t}|$, with the $j$'th entry is equal to $\sigma(\boldsymbol{\phi}^\top_j \boldsymbol{\theta}) - \sigma(\boldsymbol{\phi}^\top_j \boldsymbol{\theta}_t)$. Since we do not have access to old features $\boldsymbol{\phi}_j$, it is impossible to find the optimal $w^*_{k|t}$ for all $\boldsymbol{\theta}$ values. In what follows, we propose a practical method to estimate such compact memory for continual logistic regression, for both binary and multi-class cases. We do not aim to exactly obtain the optimal memory but still hope to keep the memory size small enough so that it is not much larger than the rank of the feature matrix. Our method uses Hessian matching which is related to other theoretical works [Li et al., 2023, Ding et al., 2024, Li et al., 2025].

## 3   Compact Memory for K-prior

We now present a practical alternative to estimate the optimal memory for continual logistic regression. Throughout, we assume a quadratic regularizer $\tfrac{1}{2}\delta_t\|\boldsymbol{\theta}\|^2$ needed for training at task $t$; the method can easily handle other regularizers as well. Motivated by Eq. 8, we propose to define the K-prior by using a set of *unit-norm* memory vectors $\mathbf{u}_{k|t}$ and their weights $w_{k|t}$, as shown below:

$$\mathcal{K}_t(\boldsymbol{\theta}; \mathbf{U}_t, \mathbf{w}_t) = \tfrac{1}{2}\delta_t\|\boldsymbol{\theta} - \boldsymbol{\theta}_t\|^2 + \sum_{k=1}^{K_t} w_{k|t}\, \mathcal{L}\left(\hat{y}(\mathbf{u}^\top_{k|t}\boldsymbol{\theta}_t)\,,\, \hat{y}(\mathbf{u}^\top_{k|t}\boldsymbol{\theta})\right), \tag{10}$$

The set of memory and weight vectors are respectively defined as follows,

$$\mathbf{U}_t = \begin{pmatrix} \mathbf{u}_{1|t}\ \mathbf{u}_{2|t}\ \ldots\ \mathbf{u}_{K_t|t} \end{pmatrix} \quad \text{and} \quad \mathbf{w}_t = \begin{pmatrix} w_{1|t}\ w_{2|t}\ \ldots\ w_{K_t|t} \end{pmatrix}, \tag{11}$$

and we denote by $\mathbf{W}_t$ the diagonal matrix whose diagonal is the vector $\mathbf{w}_t$. Clearly, a fixed $\mathbf{w}_t$ will not yield the optimal K-prior but we still hope that it will make the memory more compact. To estimate the memory and weights, we start with an initial value of $(\mathbf{U}_0, \mathbf{w}_0)$ at $t = 0$ and use the following iterative procedure (illustrated in Fig. 1),

1. Estimate $\boldsymbol{\theta}_{t+1}$ by training over $\bar{\ell}_{t+1}(\boldsymbol{\theta}; \mathcal{D}_{t+1}) + \mathcal{K}_t(\boldsymbol{\theta}; \mathbf{U}_t, \mathbf{w}_t)$.
2. Estimate $(\mathbf{U}_{t+1}, \mathbf{w}_{t+1})$ by using Hessian matching (with PPCA).

The first step is simply the standard K-prior training as in Eq. 6. Next, we motivate the second step on a linear regression problem and then present an extension for logistic regression.

### 3.1   Compact Memory for Linear Regression

We consider the following linear regression problem and show that estimating the second step can be realized via Hessian matching,

$$\sum_{j=0}^{t} \bar{\ell}_j(\boldsymbol{\theta}; \mathcal{D}_j) = \tfrac{1}{2}\delta_t\|\boldsymbol{\theta}\|^2 + \sum_{j=1}^{t}\sum_{i\in\mathcal{D}_j} \mathcal{L}\left(y_i, \hat{y}(\boldsymbol{\phi}^\top_i\boldsymbol{\theta})\right), \quad \text{where } \mathcal{L}(y, \hat{y}) = \tfrac{1}{2}(y - \hat{y})^2. \tag{12}$$

Here, we denote the quadratic regularizer $\frac{1}{2}\delta_t\|\boldsymbol{\theta}\|^2$ by $\bar{\ell}(\boldsymbol{\theta}; \mathcal{D}_0)$. For this problem, the K-prior can perfectly recover the joint loss if memory is chosen optimally, as stated in the following theorem.

**Theorem 1** *For the problem in Eq. 12, the K-prior shown in Eq. 10 is equivalent to the original loss and takes a quadratic form if we set $\boldsymbol{\theta}_t \leftarrow \boldsymbol{\theta}_t^*$, $\mathbf{U}_t \leftarrow \mathbf{U}_t^*$ and $\mathbf{W}_t \leftarrow \mathbf{S}_t^*$, that is, we have*

$$\sum_{j=0}^{t} \bar{\ell}_j(\boldsymbol{\theta}; \mathcal{D}_j) = \mathcal{K}_t(\boldsymbol{\theta}; \mathbf{U}_t, \mathbf{w}_t) = \tfrac{1}{2}(\boldsymbol{\theta} - \boldsymbol{\theta}_t)^\top \mathbf{H}(\boldsymbol{\theta}_t)(\boldsymbol{\theta} - \boldsymbol{\theta}_t), \tag{13}$$

*where $\mathbf{H}(\boldsymbol{\theta}_t) = \boldsymbol{\Phi}_{1:t}\boldsymbol{\Phi}_{1:t}^\top + \delta_t\mathbf{I}$ is the Hessian of the objective in Eq. 12.*

A proof is included in App. A.1.

This result gives a clue on how to estimate the next $(\mathbf{U}_{t+1}, \mathbf{w}_{t+1})$. Essentially, if $\boldsymbol{\theta}_{t+1} \leftarrow \boldsymbol{\theta}_{t+1}^*$ as well, then we should choose $(\mathbf{U}_{t+1}, \mathbf{w}_{t+1})$ such that the following holds for all $\boldsymbol{\theta}$,

$$\mathcal{K}_{t+1}(\boldsymbol{\theta}; \mathbf{U}_{t+1}, \mathbf{w}_{t+1}) = \bar{\ell}_{t+1}(\boldsymbol{\theta}; \mathcal{D}_{t+1}) + \mathcal{K}_t(\boldsymbol{\theta}; \mathbf{U}_t, \mathbf{w}_t). \tag{14}$$

Such a choice will ensure that the batch training loss for tasks 1 to $t+1$ is perfectly reconstructed with the new K-prior $\mathcal{K}_{t+1}$. For linear regression, this condition is satisfied by using the second-order optimality condition, obtained by taking the second derivative of both sides at $\boldsymbol{\theta}_{t+1}$. This is because the Hessian is independent of $\boldsymbol{\theta}$ and Hessian matching yields

$$\mathbf{U}_{t+1}\mathbf{W}_{t+1}\mathbf{U}_{t+1}^\top + \epsilon_{t+1}\mathbf{I} = \boldsymbol{\Phi}_{t+1}\boldsymbol{\Phi}_{t+1}^\top + \mathbf{U}_t\mathbf{W}_t\mathbf{U}_t^\top, \tag{15}$$

where $\epsilon_{t+1} = (\delta_{t+1} - \delta_t)$. We note that this is not possible by using the first-order optimality condition because gradients are of the same size as $\boldsymbol{\theta}$, which is not enough to estimate a (larger) $\mathbf{U}_t$. In contrast, the condition above is sufficient and is just another way to formulate an online SVD procedure. Essentially, we can concatenate $(\boldsymbol{\Phi}_{t+1}, \mathbf{U}_t\mathbf{W}_t^{1/2})$ and apply SVD to it to obtain $\mathbf{U}_{t+1}$ and $\mathbf{w}_{t+1}$. The last term $\epsilon_{t+1}$ represents the error made in this process which can be driven to zero by choosing $K_{t+1}$ appropriately. SVD can however be numerically unstable and we will use an alternative based on PPCA, which is easier to implement and also have an intuitive interpretation.

## 3.2 Hessian Matching via Probabilistic PCA (PPCA)

Hessian matching can be formulated as Probabilistic PCA [Tipping and Bishop, 1999a] and conveniently solved using the Expectation Maximization (EM) algorithm. Essentially, we aim to find $\mathbf{U}_{t+1}$ whose columns can accurately predict the columns of $\boldsymbol{\Phi}_{t+1}$ and $\mathbf{U}_t$. More formally, we define the following two matrices,

$$\mathbf{T} \leftarrow \sqrt{N}(\boldsymbol{\Phi}_{t+1}, \ \mathbf{U}_t\mathbf{W}_t^{1/2}), \tag{16}$$

where $N \leftarrow K_t + |\mathcal{D}_{t+1}|$. Then, we use the following PPCA model to predict the columns $\mathbf{t}_i$ of $\mathbf{T}$,

$$\mathbf{t}_i = \mathbf{U}_{t+1}\mathbf{W}_{t+1}^{1/2}\mathbf{z}_i + \mathbf{e}_i, \quad \text{where } \mathbf{z}_i \sim \mathcal{N}(0, \mathbf{I}) \text{ and } \mathbf{e}_i \sim \mathcal{N}(0, \epsilon_{t+1}\mathbf{I}). \tag{17}$$

We now show that maximum-likelihood estimation on this model can find $(\mathbf{U}_{t+1}, \mathbf{w}_{t+1})$ that solve Eq. 15. As shown by Tipping and Bishop [1999a, Eq. 4], the log marginal-likelihood of this model is obtained by noting that $\mathbf{t}_i$ follows a Gaussian distribution too, which gives (details in App. A.2)

$$\log p(\mathbf{T}|\mathbf{U}_{t+1}, \mathbf{w}_{t+1}) = -\tfrac{1}{2}N\left[\log|\mathbf{C}| + \text{Tr}\left(\mathbf{C}^{-1}\mathbf{T}\mathbf{T}^\top/N\right)\right] + \text{cnst.} \tag{18}$$

where $\mathbf{C} = \mathbf{U}_{t+1}\mathbf{W}_{t+1}\mathbf{U}_{t+1}^\top + \epsilon_{t+1}\mathbf{I}$. Then, taking the derivative and setting it to 0, we can show that the optimality condition is to set $\mathbf{C} = \mathbf{T}\mathbf{T}^\top/N$ which is equivalent to Eq. 15; a derivation is included in App. A.2. Similarly to online SVD, maximum likelihood over PPCA also performs Hessian matching. PPCA also enables us to estimate $\epsilon_{t+1}$, but for simplicity we fix it to a constant value (denoted by $\epsilon$ from now onward).

The main advantage of this formulation is that it leads to an easy-to-implement EM algorithm. Essentially, in the E-step, we find the posterior over $\mathbf{z}_i$, then in M-step, we update $\tilde{\mathbf{U}}$. The two steps can be conveniently written in just one line [Tipping and Bishop, 1999a, Eq. 29]. This is obtained by denoting $\tilde{\mathbf{U}} \leftarrow \mathbf{U}_{t+1}\mathbf{W}_{t+1}^{1/2}$ and updating it as follows:

$$\tilde{\mathbf{U}} \leftarrow \mathbf{S}\tilde{\mathbf{U}}(\epsilon\mathbf{I} + \mathbf{M}^{-1}\tilde{\mathbf{U}}^\top\mathbf{S}\tilde{\mathbf{U}})^{-1}, \quad \text{where} \quad \mathbf{M} = \tilde{\mathbf{U}}^\top\tilde{\mathbf{U}} + \epsilon\mathbf{I} \quad \text{and} \quad \mathbf{S} = \mathbf{T}\mathbf{T}^\top/N. \tag{19}$$

| **Algorithm 1:** Hessian Matching by PPCA | **Algorithm 2:** Extension for Logistic Regression |
|---|---|
| **Require:** $\mathbf{\Phi}_{t+1}, \mathbf{U}_t, \mathbf{w}_t$ and $\epsilon$ | **Require:** $\mathbf{\Phi}_{t+1}, \mathbf{U}_t, \mathbf{w}_t, \epsilon$, and $\boldsymbol{\theta}_{t+1}$ |
| 1: $\mathbf{W} \leftarrow \mathrm{Diag}(\mathbf{w}_t)$ and $\tilde{\mathbf{U}} \leftarrow \mathbf{U}_t \mathbf{W}^{1/2}$ | 1: $\mathbf{W} \leftarrow \mathrm{Diag}(\mathbf{w}_t)$ and $\mathbf{U} \leftarrow \mathbf{U}_t$ |
| 2: | 2: $\boldsymbol{\lambda}_1 \leftarrow \hat{y}'(\mathbf{\Phi}_{t+1}^\top \boldsymbol{\theta}_{t+1})$ and $\boldsymbol{\lambda}_2 \leftarrow \hat{y}'(\mathbf{U}_t^\top \boldsymbol{\theta}_{t+1})$ |
| 3: $\mathbf{T} \leftarrow (\mathbf{\Phi}_{t+1}, \ \mathbf{U}_t \mathbf{W}^{1/2})$ | 3: $\mathbf{T} \leftarrow (\mathbf{\Phi}_{t+1} \mathrm{Diag}(\boldsymbol{\lambda}_1)^{1/2}, \ \mathbf{U}_t \mathbf{W}^{1/2} \boldsymbol{\lambda}_2^{1/2})$ |
| 4: $\mathbf{S} \leftarrow \mathbf{T}\mathbf{T}^\top$ | 4: $\mathbf{S} \leftarrow \mathbf{T}\mathbf{T}^\top$ |
| 5: **while** not converged **do** | 5: **while** not converged **do** |
| 6: | 6: $\quad \boldsymbol{\lambda} \leftarrow \hat{y}'(\mathbf{U}^\top \boldsymbol{\theta}_{t+1})$ and $\tilde{\mathbf{U}} \leftarrow \mathbf{U}(\mathbf{W}\boldsymbol{\lambda})^{1/2}$ |
| 7: $\quad \mathbf{M} \leftarrow \tilde{\mathbf{U}}^\top \tilde{\mathbf{U}} + \epsilon \mathbf{I}$ | 7: $\quad \mathbf{M} \leftarrow \tilde{\mathbf{U}}^\top \tilde{\mathbf{U}} + \epsilon \mathbf{I}$ |
| 8: $\quad \tilde{\mathbf{U}} \leftarrow \mathbf{S}\tilde{\mathbf{U}}(\epsilon \mathbf{I} + \mathbf{M}^{-1} \tilde{\mathbf{U}}^\top \mathbf{S} \tilde{\mathbf{U}})^{-1}$ | 8: $\quad \tilde{\mathbf{U}} \leftarrow \mathbf{S}\tilde{\mathbf{U}}(\epsilon \mathbf{I} + \mathbf{M}^{-1} \tilde{\mathbf{U}}^\top \mathbf{S} \tilde{\mathbf{U}})^{-1}$ |
| 9: | 9: $\quad \tilde{\mathbf{U}} \leftarrow \tilde{\mathbf{U}} \boldsymbol{\lambda}^{-1/2}$ |
| 10: **end while** | 10: $\quad \mathbf{w} \leftarrow \mathrm{diag}(\tilde{\mathbf{U}}^\top \tilde{\mathbf{U}})$ and $\mathbf{W} \leftarrow \mathrm{Diag}(\mathbf{w})$ |
| 11: $\mathbf{w} \leftarrow \mathrm{diag}(\tilde{\mathbf{U}}^\top \tilde{\mathbf{U}})$ and $\mathbf{W} \leftarrow \mathrm{Diag}(\mathbf{w})$ | 11: $\quad \mathbf{U} \leftarrow \tilde{\mathbf{U}} \mathbf{W}^{-1/2}$ |
| 12: $\mathbf{U} \leftarrow \tilde{\mathbf{U}} \mathbf{W}^{-1/2}$ | 12: **end while** |
| 13: **return** $\mathbf{U}_{t+1} \leftarrow \mathbf{U}$ and $\mathbf{w}_{t+1} \leftarrow \mathbf{w}$ | 13: **return** $\mathbf{U}_{t+1} \leftarrow \mathbf{U}$ and $\mathbf{w}_{t+1} \leftarrow \mathbf{w}$ |

Figure 2: The left side shows a Hessian-matching algorithm for linear regression obtained using EM for PPCA. On the right, an extension for logistic regression is shown where differences to the regression case highlighted in red. The main differences are in line 2, 3, 6, and 9, and they all mostly due to the inclusion of terms that contain $\hat{y}'(f)$. For logistic regression, $\hat{y}'(f) = \sigma(f)[1 - \sigma(f)]$. When operated on a vector $\mathbf{f}$, $\hat{y}'(\mathbf{f})$ returns a vector of $\hat{y}'(f_i)$. $\mathrm{Diag}(\mathbf{w})$ denotes a diagonal matrix with vector $\mathbf{w}$ as the diagonal, while $\mathrm{diag}(\mathbf{W})$ returns back the diagonal vector of $\mathbf{W}$.

Derivation is included in App. A.2 and an algorithm to implement this procedure is outlined in Alg. 1, where in lines 11-12, we normalize columns of $\tilde{\mathbf{U}}$ to obtain $(\mathbf{U}_{t+1}, \mathbf{w}_{t+1})$. The algorithm assumes that $K_{t+1} = K_t$, but if memory size is increased, then we need just one additional step to initialize the additional vectors (we can do this in line 1). In practice, we use a subset of $\mathbf{\Phi}_{t+1}$ to initialize the new memory vectors. The EM procedure implements Hessian matching by essentially finding new memory vectors that can predict old memories as well as the new features.

### 3.3 An Extension for Binary Logistic Regression

We now show an extension for binary logistic regression. The procedure also works for other generalized linear models but we omit it because the derivation is straightforward. We will however briefly discuss an extension to multi-class afterward.

For binary logistic regression, aiming for a recursion similar to Eq. 14 is challenging. As discussed earlier, this is because computing the optimal $w_{k|t}$ is infeasible in practice. To simplify the estimation, we relax this requirement and aim for Hessian matching at $\boldsymbol{\theta}_{t+1}$. That is, we reformulate the search for $(\mathbf{U}_{t+1}, \mathbf{w}_{t+1})$ as the following Hessian-matching problem at $\boldsymbol{\theta}_{t+1}$,

$$\nabla^2 \mathcal{K}_{t+1}(\boldsymbol{\theta}_{t+1}; \mathbf{U}_{t+1}, \mathbf{w}_{t+1}) = \nabla^2 \bar{\ell}_{t+1}(\boldsymbol{\theta}_{t+1}; \mathcal{D}_{t+1}) + \nabla^2 \mathcal{K}_t(\boldsymbol{\theta}_{t+1}; \mathbf{U}_t, \mathbf{w}_t). \quad (20)$$

This condition leads to an equation shown below which takes a form similar to Eq. 15 obtained for the linear regression case. A derivation is given in App. A.3, where we show that the only difference is in the diagonal matrices which are defined differently:

$$\mathbf{U}_{t+1} \tilde{\mathbf{W}}_{t+1} \mathbf{U}_{t+1}^\top + \epsilon_{t+1} \mathbf{I} = \mathbf{\Phi}_{t+1} \mathbf{B}_{t+1} \mathbf{\Phi}_{t+1}^\top + \mathbf{U}_t \tilde{\mathbf{W}}_t \mathbf{U}_t^\top, \quad (21)$$

where $\tilde{\mathbf{W}}_{t+1}, \tilde{\mathbf{W}}_t$, and $\mathbf{B}_{t+1}$ are the diagonal matrices defined as follows:

$$\tilde{\mathbf{W}}_{t+1} = \mathbf{W}_{t+1} \hat{y}'(\mathbf{U}_{t+1}^\top \boldsymbol{\theta}_{t+1}), \quad \tilde{\mathbf{W}}_t = \mathbf{W}_t \hat{y}'(\mathbf{U}_t^\top \boldsymbol{\theta}_{t+1}), \quad \mathbf{B}_{t+1} = \mathrm{Diag}[\hat{y}'(\mathbf{\Phi}_{t+1}^\top \boldsymbol{\theta}_{t+1})]. \quad (22)$$

Here, the function $\hat{y}'(f)$ denotes the first derivative of $\hat{y}(f)$ with respect to $f$. For binary logistic regression, $\hat{y}(f) = \sigma(f)$, therefore $\hat{y}'(f) = \sigma(f)(1 - \sigma(f))$. When applied to a vector $\mathbf{f}$, $\hat{y}'(\mathbf{f})$ returns a vector of $\hat{y}(f_j)$, which is the case for all the quantities used above. In last expression $\mathrm{Diag}(\mathbf{f})$ forms a diagonal matrix whose diagonal is the vector $\mathbf{f}$.

The new Hessian matching equation can also be solved using an algorithm similar to Alg. 1, shown in Alg. 2. We will now briefly describe how to derive the algorithm. First, instead of using Eq. 16, we redefine $\mathbf{T}$ as shown below where we include some vectors computed using the function $\hat{y}(\cdot)$,

$$\mathbf{T} \leftarrow \sqrt{N}(\mathbf{\Phi}_{t+1}\mathrm{Diag}(\boldsymbol{\lambda}_1)^{1/2}, \ \mathbf{U}_t\mathbf{W}_t^{1/2}\boldsymbol{\lambda}_2^{1/2}), \tag{23}$$

where we use $\boldsymbol{\lambda}_1 \leftarrow \hat{y}'(\mathbf{\Phi}_{t+1}^\top\boldsymbol{\theta}_{t+1})$ and $\boldsymbol{\lambda}_2 \leftarrow \hat{y}'(\mathbf{U}_t^\top\boldsymbol{\theta}_{t+1})$. With this change, $\mathbf{TT}^\top/N$ is equal to the matrix in the right hand side of Eq. 20. This is done in line 2 and 3 of Alg. 2. The next change is in the update of $\mathbf{U}_{t+1}$. Essentially, we redefine

$$\tilde{\mathbf{U}}_{t+1} \leftarrow \mathbf{U}_{t+1}\mathbf{W}_{t+1}^{1/2}\boldsymbol{\lambda}^{1/2} \tag{24}$$

where $\boldsymbol{\lambda} \leftarrow \hat{y}'(\mathbf{U}_{t+1}^\top\boldsymbol{\theta}_{t+1})$. With this change, the left hand side in Eq. 20 becomes $\tilde{\mathbf{U}}\tilde{\mathbf{U}}^\top$, and then we can apply the update shown in Eq. 19. This is done in line 6-8. Finally, because we need to recompute $\boldsymbol{\lambda}$ every time we update $\tilde{\mathbf{U}}$, we regularly extract $(\mathbf{U}_{t+1}, \mathbf{w}_{t+1})$ from $\tilde{\mathbf{U}}_{t+1}$, which is done in line 9-12. One minor change is to initialize the procedure in line 1.

The computational cost of the EM algorithm is dominated by the inversion in line 8 of a square matrix of size $K_{t+1}$. Therefore, the overall complexity is in $\mathcal{O}(IK_{t+1}^3)$ where $I$ is the number of iterations. For small $K_{t+1}$ and $I$, this cost is manageable and does not add a huge overhead to overall training.

## 3.4 Extensions for Multi-Class Logistic Regression and Other Generalized Linear Models

Alg. 2 can be easily extended to estimate compact memory for other generalized linear models. The main change required is to appropriately define the function $\hat{y}'(f)$. We demonstrate this for multi-class logistic regression with number of classes $C$. For this case, we define a parameter vector $\boldsymbol{\theta}^{(c)}$ for each class $c$, giving rise to the matrix $\mathbf{\Theta} = (\boldsymbol{\theta}^{(1)} \ldots \boldsymbol{\theta}^{(C)})^\top$. The linear predictor for a feature $\boldsymbol{\phi}$ (of length P) is defined as $\mathbf{f} = \mathbf{\Theta}\boldsymbol{\phi}$ and is of length $C$. Given such a linear predictor, the prediction function and its derivatives are defined through the softmax function as shown below,

$$\hat{y}(\mathbf{f}) = \mathbf{p} \ \text{ and } \ \hat{y}'(\mathbf{f}) = \mathrm{Diag}(\mathbf{p}) - \mathbf{pp}^\top, \ \text{where } c\text{'th entry of } \mathbf{p} \text{ is } p^{(c)} = \frac{\exp(f^{(c)})}{\sum_{c'=1}^C \exp(f^{(c')})}, \tag{25}$$

which are vector and matrix of size $P$, respectively. The gradient and Hessian are given by

$$\frac{\partial \mathcal{L}(\mathbf{y}, \hat{y}(\mathbf{f}))}{\partial \boldsymbol{\theta}^{(c)}} = [\hat{y}(f^{(c)}) - y^{(c)}]\boldsymbol{\phi}, \qquad \frac{\partial^2 \mathcal{L}(\mathbf{y}, \hat{y}(\mathbf{f}))}{\partial \boldsymbol{\theta}^{(c)}\partial \boldsymbol{\theta}^{(c')\top}} = [\hat{y}'(\mathbf{f})]^{(c,c')}\boldsymbol{\phi}\boldsymbol{\phi}^\top,$$

where $[\hat{y}'(\mathbf{f})]^{(c,c')}$ denotes the entry $(c,c')$ of the matrix. As a result, the size of the Hessian of the loss $\mathcal{L}(y, \hat{y}(\mathbf{f}))$ with respect to (vectorized) $\mathbf{\Theta}$ is a $PC \times PC$ matrix where $P$ is the length of each $\boldsymbol{\theta}^{(c)}$. Since a $P \times P$ matrix is sufficient to estimate the memory vectors of size $P \times K_{t+1}$, we choose to perform Hessian matching over the sum $\sum_{c=1}^C \frac{\partial^2 \mathcal{L}}{\partial \boldsymbol{\theta}^{(c)}\partial \boldsymbol{\theta}^{(c)\top}}$, that is, we use the following,

$$\sum_{c=1}^C \nabla^2 \mathcal{K}_{t+1}(\boldsymbol{\theta}_{t+1}^{(c)}; \mathbf{U}_{t+1}, \mathbf{w}_{t+1}) = \sum_{c=1}^C \left[ \nabla^2 \bar{\ell}_{t+1}(\boldsymbol{\theta}_{t+1}^{(c)}; \mathcal{D}_{t+1}) + \nabla^2 \mathcal{K}_t(\boldsymbol{\theta}_{t+1}^{(c)}; \mathbf{U}_t, \mathbf{w}_t) \right]. \tag{26}$$

This condition ignores the cross-derivatives between all pairs $c \neq c'$, but the Hessian matching takes a similar form to Eq. 21. Essentially, we can push the sum with respect to $c$ inside because the only quantity that depends on $c$ is $\hat{y}(f^{(c)})$. Due to this change, we need to redefine the diagonal entries of the matrices $\tilde{\mathbf{W}}_{t+1}, \tilde{\mathbf{W}}_t$, and $\mathbf{B}_{t+1}$ by using the trace of $\hat{y}'(\mathbf{f})$, as shown below (details in App. A.4):

$$\begin{aligned}
\tilde{\mathbf{W}}_{t+1}^{(kk)} &= w_{k|t+1}\mathrm{Tr}\left[\hat{y}'(\mathbf{\Theta}_{t+1}\mathbf{u}_{k|t+1})\right], \qquad \tilde{\mathbf{W}}_t^{(kk)} = w_{k|t}\mathrm{Tr}\left[\hat{y}'(\mathbf{\Theta}_{t+1}\mathbf{u}_{k|t})\right], \\
\mathbf{B}_{t+1}^{(jj)} &= \mathrm{Tr}\left[\hat{y}'(\mathbf{\Theta}_{t+1}\boldsymbol{\phi}_j)\right],
\end{aligned} \tag{27}$$

where $k$ runs from 1 to $K_t$ (or $K_{t+1}$) and $j$ runs from 1 to $|\mathcal{D}_{1+t}|$. The trace function is denoted by $\mathrm{Tr}(\hat{y}'(\mathbf{f})) = \sum_c p^{(c)}(1 - p^{(c)})$ which takes a similar form to the binary case. Essentially, a large value of this quantity indicates that the corresponding feature is important for at least one class. To improve numerical stability, we clip the vector $\mathbf{p}$ away from 1 or 0 by a small amount (we use $10^{-4}$). Overall, this example demonstrates how to extend our approach to other generalized linear models.

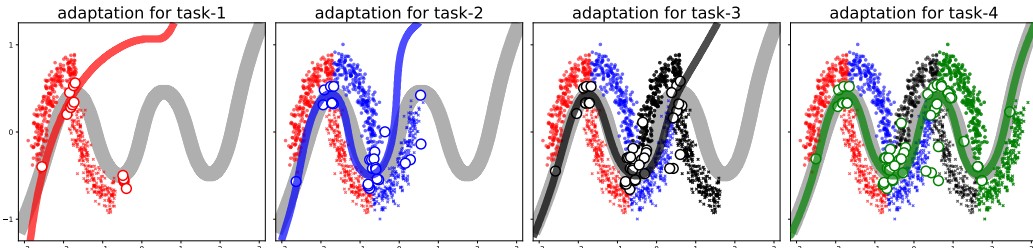

Figure 3: Binary logistic regression on a toy 'four-moon' datasets with four tasks (red → blue → black→ green). The gray line shows the batch training, while our compact-memory method's boundary are shown with colors. We see that estimated boundary closely tracks the batch solution and ultimately recovers it at the end of training. We also show with circles the points in the input space that are closed to the memories in terms of the loss gradient.

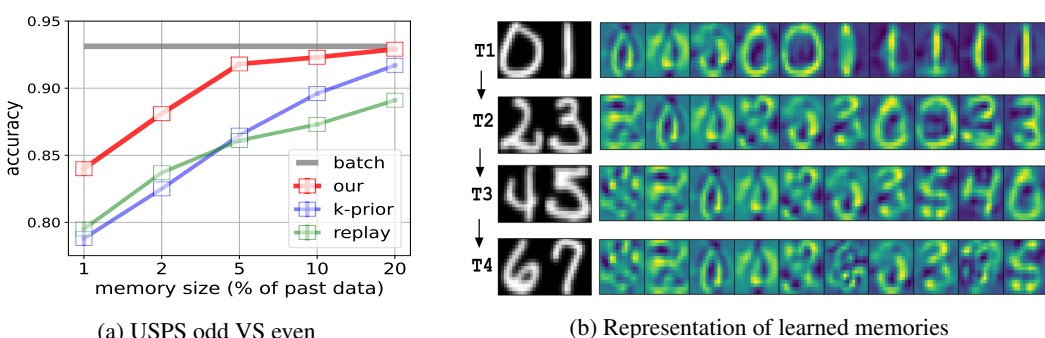

(a) USPS odd VS even  (b) Representation of learned memories

Figure 4: Panel (a) shows results for binary classification on USPS-odd-vs-even dataset. For each $\mathbf{x} \in \mathbb{R}^{256}$, the 1-order polynomial feature-map is used, that is, $\boldsymbol{\phi}(\mathbf{x}) = [1, \mathbf{x}] \in \mathbb{R}^{257}$ is used. Our method clearly beats both replay and K-prior (no learned memories) and obtains performance close to batch. Panel (b) visualizes the learned memories when using memory size of $1\%$. Each row shows the learned memory (as images) sorted according to their weights $w_{k|t}$ (left to right in descending order). We clearly see that new features emerge as new tasks arrive.

## 4 Experimental Results

We show the following results: (i) multi-output linear regression for sanity check, (ii) two binary logistic regression on toy data and USPS dataset, and (iii) multi-class logistic regression on four datasets based on CIFAR-10, CIFAR-100, TinyImageNet-200, and ImageNet-100. The multi-output regression result is included in App. B.1 where we compare to the vanilla SVD and show that our method is numerically more robust. The rest of the results are described in the next section. More details of the experiments are also included in App. B.

### 4.1 Binary Logistic Regression

We show that for continual logistic regression our method improves over the replay and vanilla K-prior methods. We start with the toy example on the 'four-moon' dataset and then show results for the USPS dataset. Both results follow the experimental setting used by Khan and Swaroop [2021].

**Four-moon dataset.** We split the four-moon dataset into four binary tasks and perform continual logistic regression over them. The training set of each task consists of 500 data points and we test over all inputs obtained on a 2-D grid $(-3.2, 3.2) \times (-1.2 \times 1.2)$ in the input space (a total of 158,632). We use a polynomial feature map of degree 5. The memory size is set to the half of feature dimensions per task and increased in this fashion with every new task. Fig. 3 shows the changes in the decision boundary as the model trains on tasks, where we see that the classifier closely follows the decision boundary of the batch training (shown in gray), ultimately recovering them after 4 tasks.

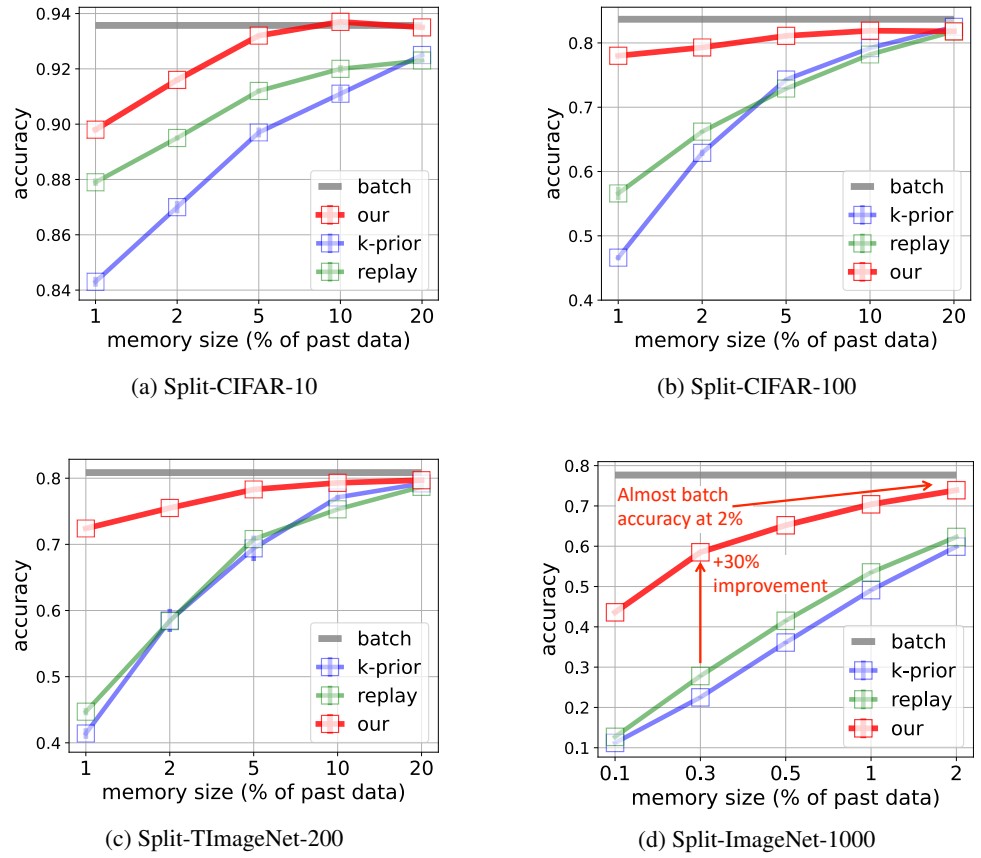

(a) Split-CIFAR-10

(b) Split-CIFAR-100

(c) Split-TImageNet-200

(d) Split-ImageNet-1000

Figure 5: Results for multi-class logistic regression. For panel (a), we use features extracted using a pre-trained Vit-B/32 and the rest use features of a pre-trained Vit-L/14. Similarly to binary classification, our method get much better accuracy compared to replay and K-prior for the same memory size. For instance, on Split-ImageNet-1000, we get $30\%$ improvement over replay when using memory size equivalent to $0.3\%$ of the data size. With $2\%$ memory, we obtain $74\%$ accuracy which is close to the batch accuracy of $77.6\%$ on this dataset.

To understand the nature of the newly learned memories, we also plot the data examples that are closest to the learned memories (shown with circles in Fig. 3). To be precise, we find examples $\mathbf{x}_*$ in the input space whose gradient is closest to a memory vector, that is, for a given $\mathbf{u}_{k|t}$, we look for $\mathbf{x}$ with the smallest $\|\nabla_{\boldsymbol{\theta}}\mathcal{L}(y, \hat{y}(f_{\boldsymbol{\theta}}(\mathbf{u}_{k|t}))) - \nabla_{\boldsymbol{\theta}}\mathcal{L}(y, \hat{y}(f_{\boldsymbol{\theta}}(\mathbf{x})))\|_2$ at $\boldsymbol{\theta} = \boldsymbol{\theta}_t$. We see that learned memories cover areas in the input space close to the decision boundary. This confirms that the memories are a reasonable representation of the decision boundary.

**USPS odd vs even datasets.** The USPS dataset is a $16\times16$ image dataset of digits from 0 to 9. We relabel each digit based on whether it is even or odd and consider a continual logistic regression with 5 task sequences given as $(0, 1)\rightarrow(2, 3)\rightarrow(4, 5)\rightarrow(6, 7)\rightarrow(8, 9)$. The training set of each task consists of 1000 data points, and the test set of each task consists of 300 data points. We use degree-1 polynomial feature-map yielding 256-dimensional features.

We consider the following baselines: K-prior [Khan and Swaroop, 2021] and experience replay [Rolnick et al., 2019] that store partial datasets of past tasks as memory. The replay set of K-prior and experience replay are initialized by random data subsets as done in Daxberger et al. [2023]. We also compare to the batch training.

Fig. 4a shows the averaged accuracies on the test set over five tasks when the varying memory sizes are considered. For each experiment we use five different random seeds. We plot results for memory sizes equivalent to 1, 2, and $5\%$ of dataset for memory. The result shows that our method outperforms other baselines significantly when using small memory sizes. Fig. 4b shows the visualization of the

ten trained memories when memory size is set to $1\%$ of the data. The memory vectors are sorted according to their weights $w_{k|t}$ (left to right in descending order) and visualized in an image form. With new tasks arriving, memories with new features are learned.

## 4.2 Multi-Class Logistic Regression using Pre-Trained Neural Network Features

We now show results following those done by Carta et al. [2023]. We use the feature map $\phi(x)$ of Vision Transformer (Vit) [Dosovitskiy et al., 2021] which is pre-trained using CLIP (available at `https://github.com/openai/CLIP`). These are known to yield effective features for classifying various image datasets [Radford et al., 2021]. We construct the features $\{\phi(x); (x, y) \in \mathcal{D}_t\}$ for $t$-th task $\mathcal{D}_t$ while freezing the parameters of the feature extractor and then use them for multi-class classification with logistic regression. We use the same baselines as the previous experiment.

We compare the performance on the following benchmarks:

1. Split-CIFAR-10 is a sequence of 5 tasks constructed by dividing 50,000 examples of CIFAR-10 into non-overlapping subsets. Each task contains 2consecutive classes.
2. Split-CIFAR-100 is a sequence of 20 tasks constructed by dividing 50,000 examples of CIFAR-100 into non-overlapping subsets. Each task contains 5 consecutive classes.
3. Split-TinyImageNet-200 is a sequence of 20 tasks constructed by dividing 100,000 examples of TinyImageNet-200 into non-overlapping subsets. Each task contains 10 consecutive classes.
4. Split-ImageNet-1000 is a sequence of 10 tasks constructed by dividing 1,281,167 examples of ImageNet-1000 into non-overlapping subsets. Each task contains 100 consecutive classes.

**Results.** Fig. 5 shows the test accuracy on each benchmark over 5 random seeds. In each figure, the $x$-axis shows the memory size and the $y$-axis shows the average accuracy over all tasks. Notably, when only $1\%$ of the dataset is used for memory, our method significantly outperforms both replay and K-prior that use randomly selected memory. For instance, on Split-ImageNet-1000, we get $30\%$ improvement by just using $0.3\%$ memory size, and obtain close to batch performance by just using $2\%$ of the data ($74\%$ vs $77.6\%$ obtained by batch). These results clearly show the memory efficiency of our method.

Fig. 6 illustrates the computational overhead of our method compared to experience replay. Under the same setting as the Split-ImageNet-1000 experiment, we consider 5 and 10 EM iterations per task; the result of 10 iterations are reported in Fig. 5. The ratio of running times, ours to replay, is reported to directly compare computational overhead. The result shows that the computational overhead of our method increases only marginally when using a smaller amount of memory. We believe that this level of computational overhead is acceptable for learning compact memory, which is the primary target of our research.

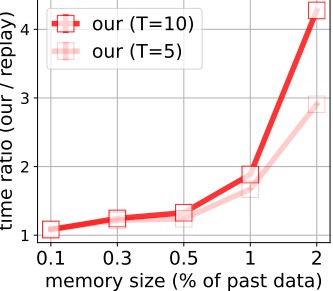

Figure 6: Running time.

## 5 Discussion

As large pre-trained neural networks show remarkable performance, adaptation of these models to new environments is becoming increasingly important. However, even in the case of continual learning with a shallow neural network, the performance tends to degrade due to catastrophic forgetting issues. To remedy this issue, models require storing a substantial memory as the learning progresses, which makes continual learning less practical.

In this work, we present a Hessian matching method to obtain compact memory for continual logistic regression. We demonstrate that, this compact memory, seeking for accurate gradient reconstructions of past tasks' losses, significantly boosts memory efficiency. We further show that the memory efficiency holds even when the compact memory is used jointly with the pre-trained neural network as a feature extractor. Although our algorithm is currently limited to the use of only the last-layer of a neural network with a frozen feature extractor, to ensure the principle of gradient reconstruction, we expect that this simple solution opens a new path to finding the compact memory for large-scale deep neural networks such as a foundation model, and that this compact memory enables these models to continually adapt to new tasks in a memory-efficient way.

## Acknowledgments and Disclosure of Funding

Y. J., T. M., and M. E. K. are supported by the Bayes duality project, JST CREST Grant Number JPMJCR2112. H. L. is supported by the Institute of Information & Communications Technology Planning & Evaluation (IITP) grant funded by the Korea government (MSIT) (No.RS-2025-02219317, AI Star Fellowship (Kookmin University)). H. L. is also supported by the Institute of Information & Communications Technology Planning & Evaluation(IITP) grant funded by the Korea government(MSIT) (No.RS-2025-02263754, Human-Centric Embodied AI Agents with Autonomous Decision-Making).

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

# A  Method Details

## A.1  Proof of Theorem 1

We first write the quadratic form by noting that $\boldsymbol{\Phi}_{1:t}\boldsymbol{\Phi}_{1:t}^\top = \mathbf{U}_t\mathbf{W}_t\mathbf{U}_t^\top$,

$$
\begin{aligned}
\mathcal{K}_t(\boldsymbol{\theta}; \mathbf{U}_t, \mathbf{w}_t) &= \tfrac{1}{2}\delta_t\|\boldsymbol{\theta} - \boldsymbol{\theta}_t\|^2 + \sum_{k=1}^{K_t} \tfrac{1}{2}w_{k|t}\,(\mathbf{u}_{k|t}^\top\boldsymbol{\theta}_t - \mathbf{u}_{k|t}^\top\boldsymbol{\theta})^2 \\
&= \tfrac{1}{2}(\boldsymbol{\theta} - \boldsymbol{\theta}_t)^\top\left(\delta_t\mathbf{I} + \mathbf{U}_t\mathbf{W}_t\mathbf{U}_t^\top\right)(\boldsymbol{\theta} - \boldsymbol{\theta}_t) \qquad (28) \\
&= \tfrac{1}{2}(\boldsymbol{\theta} - \boldsymbol{\theta}_t)^\top\left(\delta_t\mathbf{I} + \boldsymbol{\Phi}_{1:t}\boldsymbol{\Phi}_{1:t}^\top\right)(\boldsymbol{\theta} - \boldsymbol{\theta}_t) \\
&= \tfrac{1}{2}(\boldsymbol{\theta} - \boldsymbol{\theta}_t)^\top\mathbf{H}(\boldsymbol{\theta}_t)(\boldsymbol{\theta} - \boldsymbol{\theta}_t)
\end{aligned}
$$

Next, we note that $\boldsymbol{\theta}_t = \mathbf{H}(\boldsymbol{\theta}_t)^{-1}\boldsymbol{\Phi}_{1:t}^\top\mathbf{y}_{1:t}$, therefore, we can write

$$
\begin{aligned}
\mathcal{K}_t(\boldsymbol{\theta}; \mathbf{U}_t, \mathbf{w}_t) &= \tfrac{1}{2}(\boldsymbol{\theta} - \boldsymbol{\theta}_t)^\top\mathbf{H}(\boldsymbol{\theta}_t)(\boldsymbol{\theta} - \boldsymbol{\theta}_t) \\
&= \tfrac{1}{2}\boldsymbol{\theta}^\top\mathbf{H}(\boldsymbol{\theta}_t)\boldsymbol{\theta} - \boldsymbol{\theta}^\top\mathbf{H}(\boldsymbol{\theta}_t)\boldsymbol{\theta}_t + \text{cnst.} \\
&= \tfrac{1}{2}\delta_t\|\boldsymbol{\theta}\|^2 + \tfrac{1}{2}\boldsymbol{\theta}^\top\boldsymbol{\Phi}_{1:t}\boldsymbol{\Phi}_{1:t}^\top\boldsymbol{\theta} - \boldsymbol{\theta}^\top\boldsymbol{\Phi}_{1:t}^\top\mathbf{y}_{1:t} + \text{cnst.} \\
&= \tfrac{1}{2}\delta_t\|\boldsymbol{\theta}\|^2 + \sum_{j=1}^{t}\sum_{i\in\mathcal{D}_j}\tfrac{1}{2}(y_i - \boldsymbol{\phi}_k^\top\boldsymbol{\theta})^2 + \text{cnst.} \qquad (29) \\
&= \sum_{j=0}^{t}\bar{\ell}_j(\boldsymbol{\theta}; \mathcal{D}_j)
\end{aligned}
$$

This completes the proof.

## A.2  Details of PPCA and EM algorithm

The marginal likelihood in Eq. 18 is obtained by noting that, if we marginalize out $\mathbf{z}_i$ and $\mathbf{e}_i$, then $\mathbf{t}_i \sim \mathcal{N}(0, \mathbf{C})$. Since all $\mathbf{t}_i$ are identically distributed, the likelihood is simply a sum:

$$
\begin{aligned}
\log p(\mathbf{T}|\mathbf{U}_{t+1}, \mathbf{W}_{t+1}) &= \sum_{i=1}^{N}\log\mathcal{N}(\mathbf{t}_i|0, \mathbf{C}) = -\sum_{i=1}^{N}\left[\tfrac{1}{2}\log|\mathbf{C}| + \tfrac{1}{2}\mathbf{t}_i^\top\mathbf{C}^{-1}\mathbf{t}_i\right] + \text{cnst.} \\
&= -\tfrac{1}{2}\left[N\log|\mathbf{C}| + \text{Tr}\left(\mathbf{C}^{-1}\mathbf{T}\mathbf{T}^\top\right)\right] + \text{cnst.}
\end{aligned} \qquad (30)
$$

The stationarity condition can be derived by taking derivative with respect to $\tilde{\mathbf{U}} = \mathbf{U}_{t+1}\mathbf{W}_{t+1}^{1/2}$:

$$
(\mathbf{C}^{-1}\mathbf{T}\mathbf{T}^\top\mathbf{C}^{-1}\tilde{\mathbf{U}} - N\mathbf{C}^{-1}\tilde{\mathbf{U}}) = 0 \quad\Longrightarrow\quad \mathbf{T}\mathbf{T}^\top\mathbf{C}^{-1}\tilde{\mathbf{U}} = N\tilde{\mathbf{U}}, \quad\Longrightarrow\quad \mathbf{C} = \mathbf{T}\mathbf{T}^\top/N, \quad (31)
$$

where the last expression holds when $\tilde{\mathbf{U}}$ has full rank.

The EM algorithm is obtained by noting that the posterior of $\mathbf{z}_i$ is also a Gaussian

$$
\mathbf{z}_i|\mathbf{t}_i \sim \mathcal{N}(\mathbf{M}_{t+1}^{-1}\tilde{\mathbf{U}}_{t+1}^\top\mathbf{t}_i, \epsilon\mathbf{M}_{t+1}^{-1}), \qquad (32)
$$

where $\mathbf{M}_{t+1} = \tilde{\mathbf{U}}_{t+1}^\top\tilde{\mathbf{U}}_{t+1} + \epsilon\mathbf{I}$; see Tipping and Bishop [1999b, Eq. 6]. Using the mean and covariance of this Gaussian in the following M-step, we get

$$
\begin{aligned}
\tilde{\mathbf{U}}_{t+1} &\leftarrow \left(\sum_{i=1}^{N}\mathbf{t}_i\mathbb{E}[\mathbf{z}_i]^\top\right)\left(\sum_{i=1}^{N}\mathbb{E}[\mathbf{z}_i\mathbf{z}_i^\top]\right)^{-1} \\
&= \left(\sum_{i=1}^{N}\mathbf{t}_i\mathbf{t}_i^\top\tilde{\mathbf{U}}_{t+1}\mathbf{M}_{t+1}^{-1}\right)\left(\sum_{i=1}^{N}\left[\epsilon\mathbf{M}_{t+1}^{-1} + \mathbf{M}_{t+1}^{-1}\tilde{\mathbf{U}}_{t+1}^\top\mathbf{t}_i\mathbf{t}_i^\top\tilde{\mathbf{U}}_{t+1}\mathbf{M}_{t+1}^{-1}\right]\right)^{-1} \qquad (33) \\
&= \mathbf{S}\tilde{\mathbf{U}}_{t+1}\mathbf{M}_{t+1}^{-1}\left(\epsilon\mathbf{M}_{t+1}^{-1} + \mathbf{M}_{t+1}^{-1}\tilde{\mathbf{U}}_{t+1}^\top\mathbf{S}\tilde{\mathbf{U}}_{t+1}\mathbf{M}_{t+1}^{-1}\right)^{-1} \\
&= \mathbf{S}\tilde{\mathbf{U}}_{t+1}\left(\epsilon\mathbf{I} + \mathbf{M}_{t+1}^{-1}\tilde{\mathbf{U}}_{t+1}^\top\mathbf{S}\tilde{\mathbf{U}}_{t+1}\right)^{-1}.
\end{aligned}
$$

This is the update shown in Eq. 19.

### A.3 Details of Hessian Matching for Binary Logistic Regression

We derive the Hessian Matching equation given in Eq. 21. For binary logistic regression, the Hessian with respect to $\boldsymbol{\theta}$ is of size $P \times P$. We start by writing the expression for the Hessian of $\ell_{t+1}(\boldsymbol{\theta}; \mathcal{D}_{t+1})$,

$$\frac{\partial^2 \bar{\ell}_{t+1}(\boldsymbol{\theta}_{t+1}; \mathcal{D}_{t+1})}{\partial \boldsymbol{\theta} \partial \boldsymbol{\theta}^\top} = \sum_{j \in \mathcal{D}_{t+1}} \hat{y}'(\boldsymbol{\phi}_j^\top \boldsymbol{\theta}_{t+1}) \boldsymbol{\phi}_j \boldsymbol{\phi}_j^\top \tag{34}$$

where $\hat{y}'(f) = \sigma(f)[1 - \sigma(f)]$. The sum can be written conveniently in a $P \times P$ matrix form by defining a diagonal matrix $\mathbf{B}_{t+1}$, as shown below:

$$\boldsymbol{\Phi}_{t+1} \mathbf{B}_{t+1} \boldsymbol{\Phi}_{t+1}^\top = \boldsymbol{\Phi}_{t+1} \begin{pmatrix} \hat{y}'(\boldsymbol{\phi}_1^\top \boldsymbol{\theta}_{t+1}) & 0 & \cdots & 0 \\ 0 & \hat{y}'(\boldsymbol{\phi}_2^\top \boldsymbol{\theta}_{t+1}) & \cdots & 0 \\ \vdots & \vdots & \ddots & \vdots \\ 0 & 0 & \cdots & \hat{y}'(\boldsymbol{\phi}_N^\top \boldsymbol{\theta}_{t+1}) \end{pmatrix} \boldsymbol{\Phi}_{t+1}^\top,$$

where we denote the number of examples in $\mathcal{D}_{t+1}$ by $N$. Similarly, the Hessian of the K-prior is

$$\frac{\partial^2 \mathcal{K}_t(\boldsymbol{\theta}_{t+1}; \mathbf{U}_t, \mathbf{w}_t)}{\partial \boldsymbol{\theta} \partial \boldsymbol{\theta}^\top} = \tfrac{1}{2} \delta_t \mathbf{I}_P + \sum_{k=1}^{K_t} w_{k|t} \hat{y}'(\mathbf{u}_{k|t}^\top \boldsymbol{\theta}_{t+1}) \mathbf{u}_{k|t} \mathbf{u}_{k|t}^\top$$

$$= \tfrac{1}{2} \delta_t \mathbf{I}_P + \mathbf{U}_t \begin{pmatrix} w_{1|t} \hat{y}'(\mathbf{u}_{1|t}^\top \boldsymbol{\theta}_{t+1}) & 0 & \cdots & 0 \\ 0 & w_{2|t} \hat{y}'(\mathbf{u}_{2|t}^\top \boldsymbol{\theta}_{t+1}) & \cdots & 0 \\ \vdots & \vdots & \ddots & \vdots \\ 0 & 0 & \cdots & w_{K_t|t} \hat{y}'(\mathbf{u}_{K_t|t}^\top \boldsymbol{\theta}_{t+1}) \end{pmatrix} \mathbf{U}_t^\top. \tag{35}$$

We will denote the diagonal matrix by $\tilde{\mathbf{W}}_t$. The Hessian for the K-prior at task $t+1$ is obtained in a similar fashion. Using these Hessians, we get the condition given in Eq. 21.

### A.4 Details of Hessian Matching for Multi-Class Logistic Regression

For multi-class logistic regression, the Hessian with respect to vectorized $\boldsymbol{\Theta}$ is of size $PC \times PC$. To simplify this, we will write the expression in terms of a block corresponding to the second derivative with respect to $\boldsymbol{\theta}^{(c)}$ and $\boldsymbol{\theta}^{(c')}$ which is of size $P \times P$. In total, there are $C^2$ such blocks for all pairs of $(c, c')$, using which the full Hessian can be obtained.

As before, we start by writing the expression for the Hessian of $\ell_{t+1}(\boldsymbol{\theta}; \mathcal{D}_{t+1})$,

$$\frac{\partial^2 \bar{\ell}_{t+1}(\boldsymbol{\Theta}; \mathcal{D}_{t+1})}{\partial \boldsymbol{\theta}^{(c)} \partial \boldsymbol{\theta}^{(c')\top}} = \sum_{j \in \mathcal{D}_{t+1}} [\hat{y}'(\boldsymbol{\Theta}\boldsymbol{\phi}_j)]^{(c,c')} \boldsymbol{\phi}_j \boldsymbol{\phi}_j^\top \tag{36}$$

where $[\hat{y}'(\boldsymbol{\Theta}\boldsymbol{\phi}_j)]^{(c,c')}$ denotes the entry $(c, c')$ of the $P \times P$ matrix $\hat{y}'(\boldsymbol{\Theta}\mathbf{u}_{k|t})$. This can be written conveniently in a $P \times P$ matrix form by defining a diagonal matrix $\mathbf{B}_{t+1}^{(c,c')}$ at $\boldsymbol{\Theta}_{t+1}$, as shown below:

$$\boldsymbol{\Phi}_{t+1} \mathbf{B}_{t+1}^{(c,c')} \boldsymbol{\Phi}_{t+1}^\top = \boldsymbol{\Phi}_{t+1} \begin{pmatrix} [\hat{y}'(\boldsymbol{\Theta}_{t+1}\boldsymbol{\phi}_1)]^{(c,c')} & 0 & \cdots & 0 \\ 0 & [\hat{y}'(\boldsymbol{\Theta}_{t+1}\boldsymbol{\phi}_2)]^{(c,c')} & \cdots & 0 \\ \vdots & \vdots & \ddots & \vdots \\ 0 & 0 & \cdots & [\hat{y}'(\boldsymbol{\Theta}_{t+1}\boldsymbol{\phi}_N)]^{(c,c')} \end{pmatrix} \boldsymbol{\Phi}_{t+1}^\top.$$

Next, we write the K-prior at task $t$,

$$\mathcal{K}_t(\boldsymbol{\Theta}; \mathbf{U}_t, \mathbf{w}_t) = \sum_{c=1}^{C} \delta_t \|\boldsymbol{\theta}^{(c)} - \boldsymbol{\theta}_t^{(c)}\|^2 + \sum_{k=1}^{K_t} w_{k|t} \mathcal{L}\left(\hat{y}(\boldsymbol{\Theta}_t \mathbf{u}_{k|t}), \hat{y}(\boldsymbol{\Theta}\mathbf{u}_{k|t})\right). \tag{37}$$

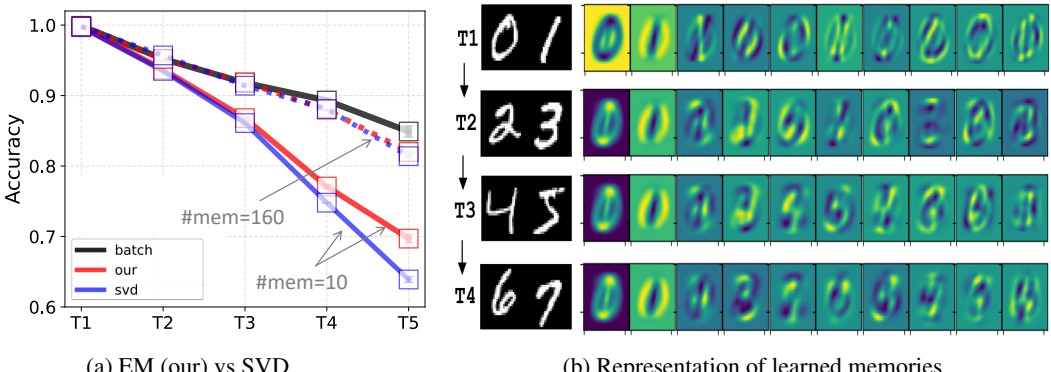

(a) EM (our) vs SVD       (b) Representation of learned memories

Figure 7: In **(a)** we compare our method with the SVD approach and batch training (oracle) on a multi-label regression task under two memory scenarios: small (10) and large (160). Our method achieves nearly the same accuracy as the SVD approach in both memory scenarios. However, for Tasks 4 and 5, our method with a small memory size outperforms the SVD approach, suggesting superior numerical robustness to gradient reconstruction errors caused by limited memory. In **(b)** we visualize the memories as more tasks are added. Each row shows the eigenimages sorted in decreasing eigenvalues from left to right. The first 2 images are consistent, and the left images with smaller eigenvalues (3rd image onward) show the merged features of digits as a new task is learned.

The Hessian is given by following (we denote the indicator by $\mathbb{I}(c = c')$ which is 1 when $c = c'$),

$$\frac{\partial^2 \mathcal{K}_t(\mathbf{\Theta}; \mathbf{U}_t, \mathbf{w}_t)}{\partial \boldsymbol{\theta}^{(c)} \partial \boldsymbol{\theta}^{(c')^\top}} = \tfrac{1}{2}\mathbb{I}(c = c')\delta_t \mathbf{I}_P + \sum_{k=1}^{K_t} w_{k|t}[\hat{y}'(\mathbf{\Theta}\mathbf{u}_{k|t})]^{(c,c')}\mathbf{u}_{k|t}\mathbf{u}_{k|t}^\top \quad = \tfrac{1}{2}\mathbb{I}(c = c')\delta_t \mathbf{I}_P$$

$$+ \mathbf{U}_t \begin{pmatrix} w_{1|t}[\hat{y}'(\mathbf{\Theta}\mathbf{u}_{1|t})]^{(c,c')} & 0 & \cdots & 0 \\ 0 & w_{2|t}[\hat{y}'(\mathbf{\Theta}\mathbf{u}_{2|t})]^{(c,c')} & \cdots & 0 \\ \vdots & \vdots & \ddots & \vdots \\ 0 & 0 & \cdots & w_{K_t|t}[\hat{y}'(\mathbf{\Theta}\mathbf{u}_{K_t|t})]^{(c,c')} \end{pmatrix} \mathbf{U}_t^\top.$$

$$(38)$$

We will denote the diagonal matrix at $\boldsymbol{\theta}_{t+1}$ by $\tilde{\mathbf{W}}_t^{(c,c')}$. The K-prior at task $t + 1$ is defined in a similar fashion. Now, applying Hessian matching, we get the following condition for all $(c, c')$ pair:

$$\mathbf{U}_{t+1}\tilde{\mathbf{W}}_{t+1}^{(c,c')}\mathbf{U}_{t+1}^\top + \mathbb{I}(c = c')\,\epsilon_{t+1}\mathbf{I} = \mathbf{\Phi}_{t+1}\mathbf{B}_{t+1}^{(c,c')}\mathbf{\Phi}_{t+1}^\top + \mathbf{U}_t\tilde{\mathbf{W}}_t^{(c,c')}\mathbf{U}_t^\top, \qquad (39)$$

There are $C^2$ such conditions, each involving a $P \times P$ matrix. Since one such condition is sufficient, We choose to match the sum over $C$ conditions for the pair $(c, c)$, that is, when $c = c'$. The advantage of this choice is that the form of Hessian matching is similar to the binary case. The only modification needed is to compute the diagonal matrices by summing of the matrix $\hat{y}'(\mathbf{f})$, which gives rise to the matrices shown in Eq. 27.

# B Experiment Details

Our code is available at https://github.com/team-approx-bayes/compact_memory_code. For all experiments, we use NVIDIA RTX 6000 Ada for experiments on Permuted-MNIST and Split-TinyImageNet. For other experiments, we use NVIDIA RTX-3090.

## B.1 Multi-Output Regression on Split-MNIST

**Experiment setting.** We consider the following hyperparameters:

- For feature map $\phi(\mathbf{x})$, we use the identity map where the raw pixel values of the image serve directly as features.

Table 1: The result on the Split-MNIST task: the performance of batch training is $0.849 \pm 0.008$.

| #Memory | 10 | 20 | 40 | 160 | 320 |
|---------|-----|-----|-----|-----|-----|
| Our | **0.697** $\pm$ 0.004 | **0.744** $\pm$ 0.001 | **0.798** $\pm$ 0.000 | **0.820** $\pm$ 0.000 | 0.815 $\pm$ 0.001 |
| SVD | 0.639 $\pm$ 0.002 | 0.685 $\pm$ 0.003 | 0.745 $\pm$ 0.001 | 0.815 $\pm$ 0.002 | **0.828** $\pm$ 0.002 |

- For hyperparameters of learning model parameter $\boldsymbol{\theta}_t$, we use Adam optimizer with learning lr $= 10^{-3}$. We use 100 epochs for each task. For the weight-space regularization hyperparameter $\delta$, we use $\delta = 0.01$.

- For hyperparameters of learning memory $\mathbf{U}_{t+1}$, we run $10,000$ iterations for each task. For the noise hyperparameter $\epsilon$ for the EM algorithm, we use $\epsilon = 10^{-3}$.

**Multi-output linear regression** We show that our EM algorithm for Hessian matching achieves compact memory. We conduct a continual multi-label regression task on **Split-MNIST** [Zenke et al., 2017], where the MNIST dataset is divided into five subsets, each corresponding to a binary classification task with label pairs (0,1), (2,3), (4,5), (6,7), and (8,9). To formulate the binary classification task as a regression problem, we map each label $k \in \{0, \ldots, 9\}$ into a vector $\mathbf{1}_k - \frac{1}{10}\mathbf{1}$ where $\mathbf{1}_k \in \mathbb{R}^{10}$ is a one-hot vector with only the $k$th component set to one and all others set to zero, and $\mathbf{1} \in \mathbb{R}^{10}$ is a vector of ones. For the feature map, we use $\phi(x) = x \in \mathbb{R}^{784}$, that is, the vectorized pixel intensities of an image. For the memory, we use constant memory size per task and accumulate its size as the model trains on new task. After training the model, we measure the classification accuracy using the entire test dataset for evaluation.

Table 1 shows the averaged accuracy of our method with the SVD approach over three seeds across memory size $\{10, 20, 40, 160, 320\}$ to investigate the memory efficiency of our method. The SVD approach obtains memory $\mathbf{U}_{t+1}$ by applying SVD to the right side of Eq.(15). Additionally, we report the result of the batch training, meaning that all tasks are trained without sequentially updating the memory and thus is regarded as the oracle performance. This result shows that our EM method is more effective than the SVD approach when using a smaller memory size (10, 20, and 40).

Fig. 7a compares our method with the SVD approach using memory sizes 10, 160, evaluated after each new task, to investigate its effectiveness with smaller memory. The x-axis denotes the task index, and the y-axis represents the average accuracy over Tasks 1 to $t$ for $t = 1, \ldots, 5$; for example, the result at T-3 indicates the average accuracy over Tasks 1–3 after training on Task 3. The results show that our method achieves nearly the same accuracy as the SVD approach up to Task 3 and outperforms it on Tasks 4 and 5. This demonstrates the robustness of our method in training later tasks, where gradient reconstruction errors are more likely to accumulate, making it harder to retain the performance on previously learned tasks. This claim is further supported by the additional results in Table 2, which show that the performance of SVD on past Tasks 1 and 2, evaluated after learning the final task, decreases substantially when a smaller memory size is used.

**Additional results.** Table 2 compares the performances of our method and the SVD approach, which is regarded as the oracle method. For memory size $K_{t+1}$, the SVD approach obtains the optimal memory

$$\mathbf{U}_{t+1} = \mathbf{U}_{1:K_{t+1}}\text{diag}(\mathbf{d}_{1:K_{t+1}}^{1/2})$$

sequentially where the columns of eigenvectors $\mathbf{U} \in \mathbb{R}^{H \times R}$ with its rank $R$ and the corresponding eigenvalues $\mathbf{d} \in \mathbb{R}_+^R$ are given by $\texttt{eigh}(\boldsymbol{\Phi}_{t+1}\boldsymbol{\Phi}_{t+1}^\top + \mathbf{U}_t\mathbf{U}_t^\top) = \mathbf{U}\text{diag}(\mathbf{d})\mathbf{U}^\top$ for the feature matrix $\boldsymbol{\Phi}_{t+1} \in \mathbb{R}^{H \times N_{t+1}}$ and the previous memory $\mathbf{U}_t \in \mathbb{R}^{H \times K_t}$. The $\Delta$**Mem** denotes the memory size allocated per task; for example, if $\Delta$**Mem** $= 10$ is used, the memory increases by 10 for each task.

This result demonstrates that our method is effective in mitigating catastrophic forgetting, especially when using a small amount of memory, because the performance on previous tasks, achieved by our method, tends to be superior to that of the SVD approach.

## B.2 Binary Logistic Regression on the Four-Moon Data Set

**Experiment setting.** We split all data points of Four-moon datasets into 4 tasks of $[500, 500, 500, 500]$ where the task proceeds as shown in Fig. 8.

Table 2: Multi-output regression on Split-MNIST over 3 runs. The performance of batch training is obtained as 0.839, comparable to the **Avg** of each method.

| $\triangle$Mem | Method | Task 1 | Task 2 | Task 3 | Task 4 | Task 5 | Avg |
|---|---|---|---|---|---|---|---|
| 10 | Ours | **0.920** $\pm$ 0.001 | **0.474** $\pm$ 0.012 | **0.616** $\pm$ 0.003 | **0.602** $\pm$ 0.001 | 0.874 $\pm$ 0.001 | **0.697** $\pm$ 0.004 |
| | SVD | 0.776 $\pm$ 0.011 | 0.337 $\pm$ 0.005 | 0.592 $\pm$ 0.006 | 0.587 $\pm$ 0.004 | **0.905** $\pm$ 0.002 | 0.639 $\pm$ 0.002 |
| 20 | Ours | **0.941** $\pm$ 0.000 | **0.566** $\pm$ 0.001 | **0.695** $\pm$ 0.002 | **0.680** $\pm$ 0.003 | 0.837 $\pm$ 0.000 | **0.744** $\pm$ 0.001 |
| | SVD | 0.831 $\pm$ 0.010 | 0.417 $\pm$ 0.006 | 0.652 $\pm$ 0.004 | 0.641 $\pm$ 0.002 | **0.884** $\pm$ 0.004 | 0.685 $\pm$ 0.003 |
| 40 | Ours | **0.959** $\pm$ 0.000 | **0.689** $\pm$ 0.001 | **0.740** $\pm$ 0.001 | **0.786** $\pm$ 0.001 | 0.816 $\pm$ 0.000 | **0.798** $\pm$ 0.000 |
| | SVD | 0.877 $\pm$ 0.006 | 0.535 $\pm$ 0.003 | 0.688 $\pm$ 0.004 | 0.735 $\pm$ 0.004 | **0.889** $\pm$ 0.001 | 0.745 $\pm$ 0.001 |
| 160 | Ours | **0.966** $\pm$ 0.000 | **0.764** $\pm$ 0.002 | 0.738 $\pm$ 0.001 | 0.848 $\pm$ 0.001 | 0.785 $\pm$ 0.000 | **0.820** $\pm$ 0.000 |
| | SVD | 0.895 $\pm$ 0.006 | 0.687 $\pm$ 0.006 | **0.744** $\pm$ 0.002 | **0.862** $\pm$ 0.003 | **0.886** $\pm$ 0.001 | 0.815 $\pm$ 0.002 |
| 320 | Ours | **0.965** $\pm$ 0.000 | **0.775** $\pm$ 0.002 | 0.720 $\pm$ 0.001 | 0.842 $\pm$ 0.001 | 0.773 $\pm$ 0.001 | 0.815 $\pm$ 0.001 |
| | SVD | 0.901 $\pm$ 0.006 | 0.730 $\pm$ 0.005 | **0.751** $\pm$ 0.004 | **0.874** $\pm$ 0.003 | **0.887** $\pm$ 0.002 | **0.828** $\pm$ 0.002 |

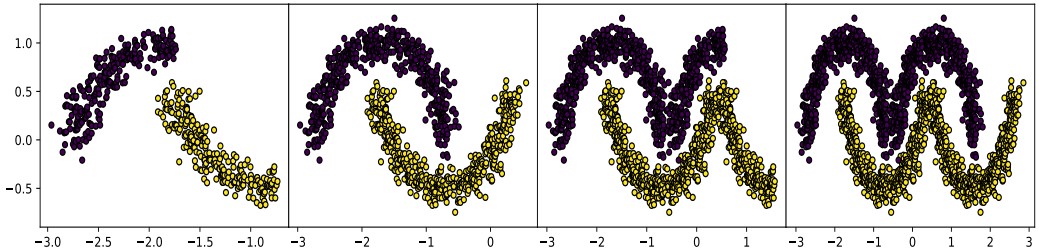

Figure 8: Sequences of four-moon classification task

We consider the following hyperparameters:

- For feature map $\phi(\mathbf{x})$, we use polynomial feature $\mathrm{poly}(5)$ yielding 21 feature dimension. We use the scikit-learn package.

- For hyperparameters of learning model parameter $\boldsymbol{\theta}_t$, we use Adam optimizer with learning $\mathrm{lr} = 0.01$. We set the number of iterations across $\{10000, 20000\}$ to learn the model parameter $\boldsymbol{\theta}_t$ for each task; we confirm the convergence of the training loss. For weight-space regularization hyperparameter $\delta$, we set $\delta = 10^{-2}$.

- For hyperparameters of learning memory $\mathbf{U}_{t+1}$, we set the number of iterations across $\{50, 100, 200\}$ for each task. For the memory size, we consider $\{7, 14, 21\}$ memory for each task and accumulate the memory as the task proceeds. For the noise hyperparameter $\epsilon$, we conduct a grid search over $\epsilon \in \{10^{-3}, 10^{-4}, 10^{-5}\}$ and use $10^{-4}$.

**Additional results across varying memory size.** Fig. 9 shows the results using the 7 and 21 memory per task. The result using 14 memory is reported in Section 4 of the main manuscript. This confirms that our method can match the decision boundary of batch training closely even when using a small amount of memory, as shown in Fig. 9a. As more memory is used, our decision boundary becomes more equal to that of the batch training, as shown in Fig. 9b.

### B.3 Binary logistic regression on USPS odd vs even dataset

**Experiment setting.** We consider the following hyperparameters:

- For feature map $\phi(\mathbf{x})$, we use polynomial feature $\mathrm{poly}(1)$ yielding 256 feature dimension. We use the scikit-learn package.

- For hyperparameters of learning model parameter $\boldsymbol{\theta}_t$, we use Adam optimizer with learning $\mathrm{lr} = 0.1$. We use 5000 iterations to learn the model parameter $\boldsymbol{\theta}_t$ for each task. For the weight-space regularization hyperparameter $\delta$, we conduct a grid search over $\delta \in \{0.5, 0.1, 0.05\}$ and use $\delta = 0.5$ for K-prior and our method.

- For hyperparameters of learning memory $\mathbf{U}_{t+1}$, we use 5000 iterations for each task. For the noise hyperparameter $\epsilon$, we conduct a grid search over $\epsilon \in \{10^{-3}, 10^{-4}, 10^{-5}\}$.

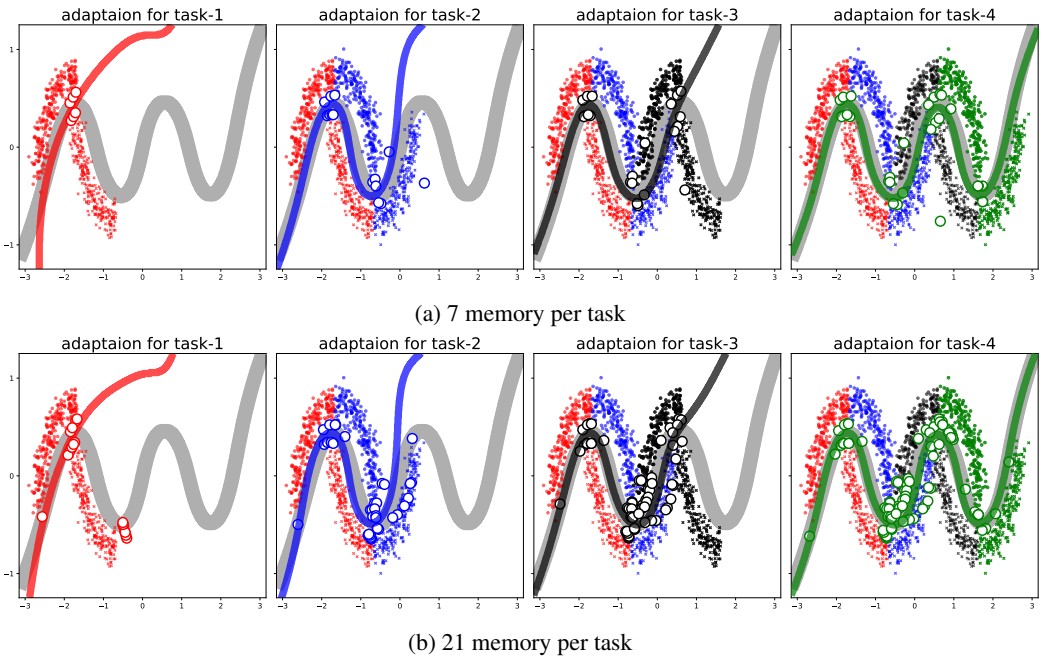

(a) 7 memory per task

(b) 21 memory per task

Figure 9: Investigation on varying memory size used for K-prior.

**Investigation on varying weight-space regularization hyperparameter for K-prior.** Fig. 10 compares K-prior and our method across the varying hyperparameters $\delta \in \{0.5, 0.1, 0.05\}$ that control the importance of weight-space regularization hyperparameter term for K-prior. This result confirms that our method improves the memory efficiency in general and is more effective at $\delta \in \{0.1, 0.5\}$ where the K-prior can match the result of batch training more closely.

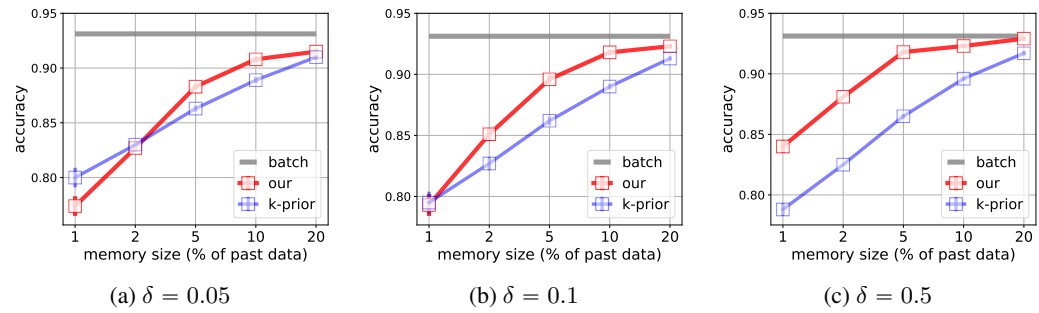

(a) $\delta = 0.05$        (b) $\delta = 0.1$        (c) $\delta = 0.5$

Figure 10: Investigation on varying hyperparameter $\delta \in \{0.05, 0.1, 0.5\}$ for K-prior

**Investigation on varying noise hyperparameter EM algorithm for PPCA.** Table 3 compares the performances of our method across varying noise hyperparameter $\epsilon$ in the EM algorithm for PPCA where we run 5 experiments with different random seeds. This confirms that our method is consistent over the noise hyperparameter $\epsilon$.

### B.4 Multi-label Logistic Regression on Split CIFAR-10, CIFAR-100, and TinyImageNet

**Experiment setting.** We consider the following hyperparameters:

- For feature map $\phi(\mathbf{x})$, we use the feature extractor of ResNet-50 and Vision Transformer (Vit) that are pretrained by CLIP. For Split CIFAR-10, we use Vit-B/32, meaning the base-scale model with 32 patch size. For Split CIFAR-100 and Split Tiny-ImageNet, we use Vit-L/14, meaning the large-scale model size with 14 patch size. The feature extractors of ResNet-50 and Vit yield 2048 features and 768 features, respectively.

Table 3: Investigation on the effect of noise hyperparameter $\epsilon \in \{10^{-3}, 10^{-4}, 10^{-5}\}$.

| $\triangle$Mem | $\epsilon$ | Task 1 | Task 2 | Task 3 | Task 4 | Task 5 | Avg |
|---|---|---|---|---|---|---|---|
| **1%** | $\epsilon = 10^{-3}$ | $0.769 \pm 0.046$ | $0.840 \pm 0.012$ | $0.566 \pm 0.033$ | $0.982 \pm 0.005$ | $0.946 \pm 0.002$ | $0.820 \pm 0.013$ |
| | $\epsilon = 10^{-4}$ | $0.791 \pm 0.029$ | $0.855 \pm 0.014$ | $0.623 \pm 0.022$ | $0.983 \pm 0.003$ | $0.952 \pm 0.005$ | $0.840 \pm 0.007$ |
| | $\epsilon = 10^{-5}$ | $0.772 \pm 0.043$ | $0.872 \pm 0.010$ | $0.627 \pm 0.032$ | $0.980 \pm 0.003$ | $0.954 \pm 0.002$ | $0.841 \pm 0.008$ |
| **2%** | $\epsilon = 10^{-3}$ | $0.860 \pm 0.010$ | $0.913 \pm 0.002$ | $0.689 \pm 0.020$ | $0.979 \pm 0.003$ | $0.934 \pm 0.003$ | $0.875 \pm 0.003$ |
| | $\epsilon = 10^{-4}$ | $0.885 \pm 0.012$ | $0.885 \pm 0.018$ | $0.724 \pm 0.008$ | $0.981 \pm 0.003$ | $0.933 \pm 0.005$ | $0.881 \pm 0.004$ |
| | $\epsilon = 10^{-5}$ | $0.891 \pm 0.006$ | $0.893 \pm 0.012$ | $0.715 \pm 0.010$ | $0.982 \pm 0.003$ | $0.937 \pm 0.002$ | $0.883 \pm 0.003$ |
| **5%** | $\epsilon = 10^{-3}$ | $0.932 \pm 0.002$ | $0.931 \pm 0.003$ | $0.810 \pm 0.011$ | $0.980 \pm 0.003$ | $0.920 \pm 0.003$ | $0.915 \pm 0.002$ |
| | $\epsilon = 10^{-4}$ | $0.950 \pm 0.002$ | $0.921 \pm 0.003$ | $0.819 \pm 0.012$ | $0.983 \pm 0.004$ | $0.915 \pm 0.005$ | $0.918 \pm 0.002$ |
| | $\epsilon = 10^{-5}$ | $0.948 \pm 0.002$ | $0.921 \pm 0.007$ | $0.798 \pm 0.009$ | $0.984 \pm 0.002$ | $0.915 \pm 0.004$ | $0.913 \pm 0.002$ |
| **10%** | $\epsilon = 10^{-3}$ | $0.933 \pm 0.002$ | $0.944 \pm 0.005$ | $0.876 \pm 0.008$ | $0.972 \pm 0.004$ | $0.878 \pm 0.003$ | $0.921 \pm 0.002$ |
| | $\epsilon = 10^{-4}$ | $0.962 \pm 0.002$ | $0.937 \pm 0.003$ | $0.855 \pm 0.009$ | $0.976 \pm 0.003$ | $0.884 \pm 0.003$ | $0.923 \pm 0.002$ |
| | $\epsilon = 10^{-5}$ | $0.969 \pm 0.003$ | $0.934 \pm 0.002$ | $0.855 \pm 0.009$ | $0.977 \pm 0.003$ | $0.889 \pm 0.003$ | $0.925 \pm 0.001$ |
| **20%** | $\epsilon = 10^{-3}$ | $0.949 \pm 0.003$ | $0.946 \pm 0.002$ | $0.891 \pm 0.005$ | $0.971 \pm 0.002$ | $0.859 \pm 0.002$ | $0.923 \pm 0.001$ |
| | $\epsilon = 10^{-4}$ | $0.979 \pm 0.001$ | $0.949 \pm 0.002$ | $0.877 \pm 0.006$ | $0.965 \pm 0.002$ | $0.872 \pm 0.003$ | $0.929 \pm 0.002$ |
| | $\epsilon = 10^{-5}$ | $0.981 \pm 0.001$ | $0.940 \pm 0.001$ | $0.868 \pm 0.006$ | $0.967 \pm 0.002$ | $0.876 \pm 0.004$ | $0.927 \pm 0.002$ |

- For hyperparameters of learning model parameter $\boldsymbol{\theta}_t$, we use Adam optimizer with learning lr $= 0.1$. We use 100 iterations for each task with 1024 batch size. For the weight-space regularization hyperparameter $\delta$, we conduct a grid search over $\delta \in \{0.1, 0.01, 0.001\}$.

- For hyperparameters of learning memory $\mathbf{U}_{t+1}$, we set 10 iterations for each task. For the noise hyperparameter $\epsilon$, we use $\epsilon = 10^{-4}$.

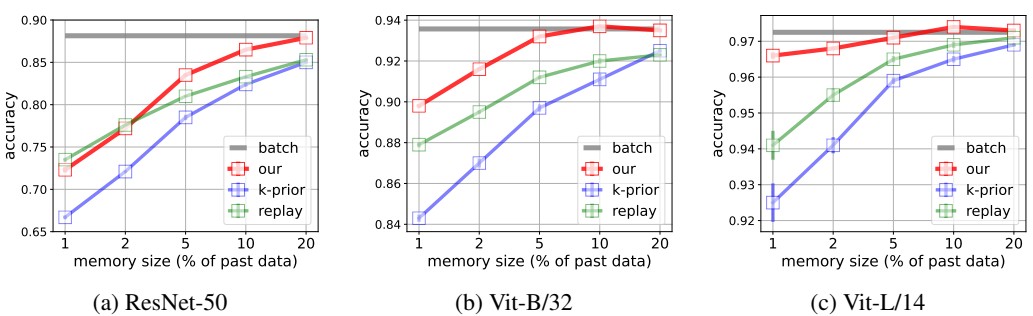

Figure 11: Investigation on varying feature extractor with Split-CIFAR-10.

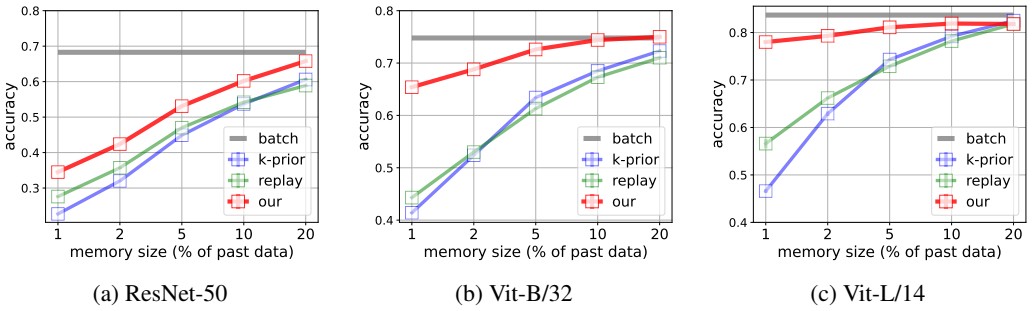

Figure 12: Investigation on varying feature extractor with Split-CIFAR-100.

**Investigation on varying feature extractor.** Fig. 11 shows the results on Split-CIFAR 10 across varying feature extractors where the feature extractors of ResNet-50, Vit-B/32, and Vit-L/14 are used in Figures 11a to 11c, respectively. Similarly, Figures 12 and 13 show the corresponding results on Split-CIFAR-100 and Split-TinyImageNet, respectively. These results confirm that our method improves memory efficiency consistently regardless of the feature extractor.

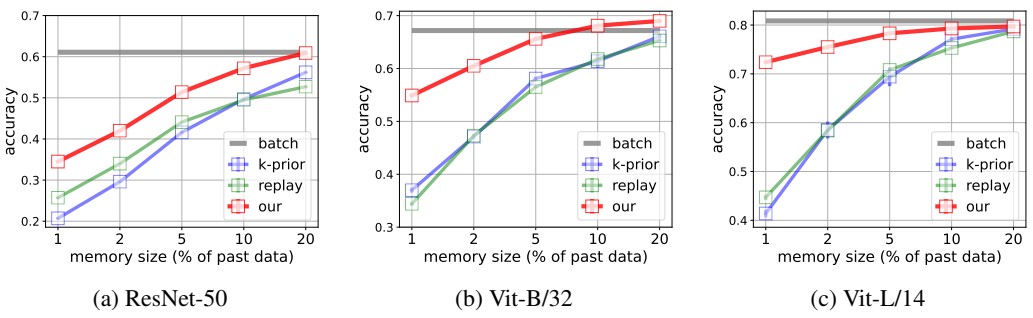

|  | (a) ResNet-50 | (b) Vit-B/32 | (c) Vit-L/14 |

Figure 13: Investigation on varying feature extractor with Split-TinyImageNet.

## B.5 Additional Results on Permuted MNIST

**Experiment setting.** Following the benchmark experiment setting in [Carta et al., 2023], we also consider Permuted-MNIST consisting of a sequence of 5 tasks constructed by permuting pixels of 60,000 training samples of MNIST for each task.

We consider the following hyperparameters:

- For feature map $\phi(\mathbf{x})$, we use the feature extractor of ResNet-50 and Vision Transformer (Vit) that are pretrained by CLIP. The feature extractor of ResNet-50 and ViT yield 2048 features and 768 features, respectively

- For hyperparameters of learning model parameter $\boldsymbol{\theta}_t$, we use Adam optimizer with learning lr = 0.1. We use 200 iterations for each task. For the weight-space regularization hyperparameter $\delta$, we conduct a grid search over $\delta \in \{0.01, 0.001\}$.

- For hyperparameters of learning memory $\mathbf{U}_{t+1}$, we set 10 iterations for each task. For the noise hyperparameter $\epsilon$, we consider $\epsilon = 10^{-4}$.

**Additional results.** Table 4 shows that our method outperforms other baselines, especially when 1, 2, and 5 percent of data points are used as memory. This result confirms that our method also improves memory efficiency on a domain incremental task.

Table 4: Performance on Permuted-MNIST across varying amount of memory.

| Feature | Method | 1% | 2% | 5% | 10% |
|---------|--------|-----|-----|-----|------|
| ResNet-50 | Replay | $0.474 \pm 0.002$ | $0.508 \pm 0.002$ | $0.554 \pm 0.001$ | $\mathbf{0.592} \pm 0.002$ |
|  | K-prior | $0.465 \pm 0.003$ | $0.493 \pm 0.002$ | $0.534 \pm 0.002$ | $0.572 \pm 0.003$ |
|  | Our | $\mathbf{0.490} \pm 0.001$ | $\mathbf{0.521} \pm 0.001$ | $\mathbf{0.561} \pm 0.002$ | $\mathbf{0.592} \pm 0.003$ |
| Vit-B/32 | Replay | $0.490 \pm 0.003$ | $0.524 \pm 0.002$ | $0.578 \pm 0.004$ | $\mathbf{0.623} \pm 0.003$ |
|  | K-prior | $0.453 \pm 0.001$ | $0.489 \pm 0.003$ | $0.549 \pm 0.005$ | $0.594 \pm 0.004$ |
|  | Our | $\mathbf{0.522} \pm 0.006$ | $\mathbf{0.545} \pm 0.007$ | $\mathbf{0.584} \pm 0.003$ | $0.613 \pm 0.004$ |

