# OpenReview forum: "Compact Memory for Continual Logistic Regression"
_NeurIPS.cc/2025/Conference — NeurIPS 2025 poster_

### Official Review · Reviewer_qXrR · 2025-06-20

**Clarity:** 3
**Significance:** 2
**Originality:** 2
**Rating:** 5
**Confidence:** 3

**Summary:**

This paper introduces a novel approach for learning compact memory in continual logistic regression. It employs a Hessian matching-based Expectation-Maximization (EM) algorithm that leverages Probabilistic Principal Component Analysis (PPCA) for K-prior regularization. This method significantly enhances memory efficiency by precisely reconstructing past task gradients. Empirical evaluations across diverse benchmarks, including those with frozen pre-trained neural network features, demonstrate superior memory efficiency and competitive accuracy against baselines, highlighting its potential for scalable continual adaptation.

**Questions:**

See weaknesses.

**Ethical Concerns:**

["NO or VERY MINOR ethics concerns only"]

**Final Justification:**

I appreciate the detailed rebuttal, which has effectively addressed my primary concerns. The new experiments on ImageNet-1k, in particular, compellingly demonstrate the method’s significant performance gains under strict memory constraints. Accordingly, I have raised my score to 5.

**Limitations:**

The limitations have been correctly discussed in Section 5.

**Paper Formatting Concerns:**

No major formatting issues were observed in the paper.

**Quality:**

3

**Strengths And Weaknesses:**

### **Strengths**

1. The paper introduces a novel, theoretically sound framework for K-prior regularization. It learns compact memory through a Hessian matching principle derived from gradient matching, solved by an EM-based PPCA algorithm.
2. A core contribution is the demonstrable improvement in K-prior's memory efficiency by learning compact memory rather than storing raw data. This is a crucial practical advancement for scaling continual learning, particularly for large-scale models.
3. The paper provides comprehensive experimental validation and extensive ablations.
4. The code is available

### **Weaknesses**

1. It is better to include larger databases such as imagenet-1k [1] into the experiments to make the results more convincing.
2. Please clarify what "compact memory" entails. Does it mean representing more sample information within the same buffer size? If so, what is the quantitative gain (e.g., how many times more information) compared to storing raw images?
3. In comparisons with Experience Replay (ER) [2], was the buffer size held constant (allowing for more represented sample information), or was the number of stored samples kept constant? This distinction is crucial for assessing the fairness of the comparison.
4. Since Nearest Class Mean (NCM) classifiers are effective and widely used for continual learning with frozen feature extractors [3,4,5,6], please justify why storing compact memory representations of *samples* (features) is preferred over maintaining single *class prototypes*, especially if NCM yields comparable or superior performance.
5. Equation (1) seems problematic. Additionally, some equations (7, 8, 9) are missing punctuation (e.g., periods or commas) at their conclusion.

[1] Daxberger, Erik, et al. "Improving continual learning by accurate gradient reconstructions of the past." *TMLR. 2023.*

[2] Rolnick, David, et al. "Experience replay for continual learning." *NeurIPS. 2019.*

[3] Zhou, Da-Wei, et al. "Revisiting class-incremental learning with pre-trained models: Generalizability and adaptivity are all you need." *IJCV. 2025.*

[4] Zhou, Da-Wei, et al. "Expandable subspace ensemble for pre-trained model-based class-incremental learning." *CVPR. 2024.*

[5] Prabhu, Ameya, et al. "Random Representations Outperform Online Continually Learned Representations." *NeurIPS. 2024.*

[6] Goswami, Dipam, et al. "Fecam: Exploiting the heterogeneity of class distributions in exemplar-free continual learning." *NeurIPS. 2023.*

---

> ### Author Rebuttal · Authors · 2025-07-30
>
> We thank the reviewer for their comments. We have tried to address them below and hope to discuss more with the reviewer. If the reviewer's concerns are addressed appropriately, it will be great if they can consider increasing their score. Thanks for reading our work!
>
> > Q1. It is better to include larger databases such as imagenet-1k [1] into the experiments to make the results more convincing.
>
> Thanks for the suggestion. We have now included ImageNet-1k where we find our method to again perform better. Following [1], we split the dataset into 10 tasks each with 100 consecutive classes and consider memory size of 0.5% (600 per task), 1% (1200 per task), and 2% (2400 per task) respectively. Below are the results:
>
> | #Memory | Replay        | K-Prior       | Ours             | Batch (Oracle) |
> |:-------:|:-------------:|:-------------:|:----------------:|:------:|
> | 0.5%    | 41.7 ± 0.2    | 36.3 ± 0.6    | **65.2 ± 0.2**    | 77.6   |
> | 1%      | 54.0 ± 0.4    | 49.4 ± 0.1    | **70.4 ± 0.0**    | 77.6   |
> | 2%      | 62.3 ± 0.1    | 59.9 ± 0.5    | **73.9 ± 0.2**    | 77.6   |
>
> ---
>
> >  Q2. Please clarify what "compact memory" entails. Does it mean representing more sample information within the same buffer size? If so, what is the quantitative gain (e.g., how many times more information) compared to storing raw images?
>
> Yes, you are absolutely right. Quantitative gain can be seen in Fig. 4 and 5, for example, for tinyImagNet, to match the performance of our compact memory of 1% we need around 5% of the data for replay and K-prior. Similar results hold for other datasets.
>
> ---
> > Q3. In comparisons with Experience Replay (ER) [2], was the buffer size held constant (allowing for more represented sample information), or was the number of stored samples kept constant? This distinction is crucial for assessing the fairness of the comparison.
>
> We set the memory size (buffer size) per task and increase it whenever the model learns a new task. For example, if a continual learning experiment consists of 5 tasks and the memory size is set to 10 per task, the total memory size increases as [10,20,30,40]. This configuration of the memory size is applied equally to both ER and our method.
>
>
>
>
>
>
> ---
> > Q4. Since Nearest Class Mean (NCM) classifiers are effective and widely used for continual learning with frozen feature extractors [3,4,5,6], please justify why storing compact memory representations of samples (features) is preferred over maintaining single class prototypes, especially if NCM yields comparable or superior performance.
>
> Though NCM has been empirically effective for continual learning, to the best of our knowledge, the rationale for how NCM yields superior performance, is not entirely clear. Our work follows the more usual trend to represent the past features in a principled way that aims to recover the gradient of past task loss using a K-prior regularizer, and this could also be considered complimentary to NCM. We are happy to learn more about reviewers point of view and add discussion on such methods in the paper.
>
> ---
> > Q5. Equation (1) seems problematic. Additionally, some equations (7, 8, 9) are missing punctuation (e.g., periods or commas) at their conclusion.
>
> Thanks for pointing this out. We will fix these mistakes in the next versions.

---

> > ### Comment · Reviewer_qXrR · 2025-08-04
> >
> > I thank the authors for answering my comments.
> >
> > I believe that the authors' responses can solve most of my concerns of this paper, and I would like to re-rate this paper to 5.

---

> > > ### Author Response · Authors · 2025-08-04
> > >
> > > Thank you for updating your evaluation of our work. If you intended to revise the score but have not yet done so, we kindly ask that you update it.

---

### Official Review · Reviewer_zWmk · 2025-06-28

**Clarity:** 4
**Significance:** 3
**Originality:** 2
**Rating:** 5
**Confidence:** 3

**Summary:**

The authors develop a method for constructing a set of unlabeled pseudo datapoints (not a subset of the training data) using an EM algorithm for Probabilistic PCA (PPCA) to represent data from past tasks in a continual learning regression setting. They show that using this learned memory for regularization results in improved memory efficiency compared to prior methods which use a subset of the training data for regularization.

**Questions:**

**Questions**
- Why is SVD worse in Figure 2 (a)? I might be missing something, but SVD should result in the _optimal_ low-rank representation of the current task's data combined with the compact memory. How can PPCA be better for a small number of data points?
- How does the cost of PPCA EM compare to other numerical linear algebra methods to compute top $K$ singular vectors and values, e.g. randomized SVD? Can you add more discussion around the choice of PPCA EM and its computational cost?
- How is convergence of EM determined? How many steps of EM are run in practice?
- Is experience replay defined anywhere?
- Are there any other relevant works on continual regression besides Khan and Swaroop [2021]? What about methods for selecting subsets of training data and other works using pseudo-data for continual learning?

**Suggestions**

I will increase my score if these suggestions are followed.
- It would be very informative to add baselines with non-random subsets of training data. This would provide some insight in how much you gain from using pseudo data points compared to a curated subset of the training data. Also, it is a fair comparison since you have some additional overhead of constructing your compact memory with PPCA EM, so you could use the same amount of compute for the selection of a subset of the training data.
- Please answer my questions above and add additional discussion in the paper where necessary.

**Minor**
- Abstract:
> Our approach can drastically reduce memory size without sacrificing performance, for instance, we can obtain 30% improvement in accuracy over Experience Replay that uses 1% of the data for memory when adapting it with the the pre-trained Vision transformer as a feature extractor.

The second half of the sentence is hard to parse and not completely clear: “that uses 1% of the data for memory” -> does this mean Experience Replay uses 1% of the training data? If yes, how much less memory does the proposed method need? “when adapting it with the the pre-trained Vision transformer as a feature extractor” -> it’s unclear how that relates to the proposed method.
- Please add references to the appendix.
- Line 45/46: “(…) the memory size can be large because $\mathcal{M}_t$ in practice is chosen to be a subset of the data.” Is the point that the cardinality of the subset is a fixed percentage of the data, so for large datasets the subset might be large as well? The same can be said about compact memory (of course the goal is the the fixed percentage per task is smaller).
- $K_t$ in line 106/107 is the rank of the input matrix, in line 117 it is the memory size which should be is probably < the rank of the input matrix?

**Ethical Concerns:**

["NO or VERY MINOR ethics concerns only"]

**Final Justification:**

The authors followed the suggestions I detailed in my review (non-random training data subset selection experiment and addressing my questions). The other reviewers had similar questions (e.g. PPCA EM vs. SVD) which the authors addressed. Hence, as promised I increased my score and recommend to accept the paper.

**Limitations:**

yes

**Quality:**

3

**Strengths And Weaknesses:**

**Strengths**
- Well-written and mostly clear.
- Conceptually elegant and simple approach (using unlabeled pseudo data memory) to an important problem.
- Sufficient details about the method (see the appendix).
- Well-designed experiments and visualizations.

**Weaknesses**
- Some more discussion of the SVD result in Figure 2 (a) (see questions) and the choice of using PPCA instead of SVD would be helpful.
- Discussion of literature about (1) other approaches that construct pseudo data (for continual learning), (2) continual learning in regression settings, and (3) training data subset selection methods is missing.
- Comparing against baselines with non-randomly chosen subsets of the training data would be more informative.
- Discussion of overall computational cost compared to the other methods would be useful.

---

> ### Author Rebuttal · Authors · 2025-07-30
>
> >  Q1. Why is SVD worse in Figure 2 (a)? I might be missing something, but SVD should result in the optimal low-rank representation of the current task's data combined with the compact memory. How can PPCA be better for a small number of data points?
>
> Thanks for the question. This is most likely due to the piling up of numerical errors in SVD. This can be clearly seen by looking at the averaged test accuracy over Task $1$ to $t$ for $t=1,..,5$ during training. The following table shows that SVD performance gets suddenly worse after Task $3$; for example, the column of Task $3$ shows the averaged test accuracy over Task $1$ to $3$. This holds for all memory sizes 10, 20, 40, and 160, especially when the smaller memory is used.
>
> | Method         | Task 1       | Task 2       | Task 3       | Task 4       | Task 5       |
> |----------------|--------------|--------------|--------------|--------------|--------------|
> | **EM-10**       | 99.8 ± 0.00  | 93.7 ± 0.00  | 86.8 ± 0.20  | 77.1 ± 0.30  | 69.7 ± 0.60  |
> | **SVD-10**      | 99.7 ± 0.00  | 93.5 ± 0.10  | 86.1 ± 0.30  | 74.8 ± 0.00  | 63.9 ± 0.40  |
> | *Gap (EM − SVD)*| **+0.1**     | **+0.2**     | **+0.7**     | **+2.3**     | **+5.8**     |
> | **EM-20**       | 99.7 ± 0.00  | 94.7 ± 0.00  | 87.8 ± 0.20  | 80.6 ± 0.20  | 74.3 ± 0.10  |
> | **SVD-20**      | 99.8 ± 0.00  | 94.7 ± 0.00  | 86.7 ± 0.30  | 77.4 ± 0.20  | 68.5 ± 0.50  |
> | *Gap (EM − SVD)*| **−0.1**     | **+0.0**     | **+1.1**     | **+3.2**     | **+5.8**     |
> | **EM-40**       | 99.8 ± 0.00  | 95.2 ± 0.00  | 89.9 ± 0.10  | 85.9 ± 0.10  | 79.8 ± 0.10  |
> | **SVD-40**      | 99.7 ± 0.00  | 95.2 ± 0.00  | 88.6 ± 0.40  | 81.7 ± 0.20  | 74.5 ± 0.10  |
> | *Gap (EM − SVD)*| **+0.1**     | **+0.0**     | **+1.3**     | **+4.2**     | **+5.3**     |
> | **EM-160**      | 99.8 ± 0.00  | 95.2 ± 0.00  | 91.9 ± 0.00  | 88.0 ± 0.00  | 81.8 ± 0.30  |
> | **SVD-160**     | 99.8 ± 0.00  | 95.7 ± 0.10  | 91.4 ± 0.20  | 88.1 ± 0.30  | 81.4 ± 0.30  |
> | *Gap (EM − SVD)*| **+0.0**     | **−0.5**     | **+0.5**     | **−0.1**     | **+0.4**     |
>
> The EM algorithm is more stable because it relies on matrix inversion where an identity matrix is added (see update of $S$ in Algorithm 1). It is possible that these issues will be fixed by using a better numerical algorithm for SVD or also by tuning the delta in Eq. 3, although the latter is tricky for the sequential setting in continual learning.
>
> ---
> >  Q2. How does the cost of PPCA EM compare to other numerical linear algebra methods to compute top  singular vectors and values, e.g. randomized SVD? Can you add more discussion around the choice of PPCA EM and its computational cost?
>
> **computation cost of PPCA EM compared to top-K SVD**:
> In our problem, SVD is applied to the concatenated matrix $[ \Phi_{t+1} , U_t ] \in R^\{P \times (T+K)\} $ with $P$ being the number of parameters, $T$ being the size of new data, and $K$ being the size of memory $U_t$. The computational complexity of SVD is $\mathcal{O}( (T+K) P K )$ for top-$K$ SVD.
>
> The computational cost of the EM algorithm is $\mathcal{O}(K^{3} )$ per iteration and is dominated by inverse of $K$-size square matrices $S$ and $(S^{-1}+ MM^{\top})$; see Algorithm 1. Since $K\ll T$ and $P$ both, this is faster if the number of iterations is reasonable.
>
> In our experiments, sometimes EM took a long number of iterations to converge in which cases the cost saving compared to SVD are not obtained, but the method was still more numerically stable. It is also true that there might be other versions of SVD that could compete with the EM but for our purpose EM worked well and we did not try to find another alternative and only compared to the standard SVD.
>
> **Discussion regarding choice of PPCA EM**: Thanks for the suggestion. We will highlight the numerically stability under accumulation of error as new tasks as the main reason for choosing EM over SVD.
>
> ---
> > Q3. How is convergence of EM determined? How many steps of EM are run in practice?
>
> We keep track the norm of $|| U_{t+1} U_{t+1}^T - U_{t+1}^{old} (U_{t+1}^{old})^T ||$ and stop when it is small enough. However, we sometimes found EM to use maximum iterations (see Appendix for the max iterations for each experiment).
>
> ---
> > Q4. Is experience replay defined anywhere?
>
> Yes, the equation in the right hand side of Eq. 2 is experience replay.
>
> ---
> > Q5. Are there any other relevant works on regression besides Khan and Swaroop [2021]? What about methods for selecting subsets of training data and other works using pseudo-data for continual learning?....... Comparing baselines with non-randomly chosen subsets of the training data would be more informative.
>
> **Regression tasks**: to the best of our knowledge, we have not found relevant work that takes a similar approach to ours, as most continual learning research has focused on image classification task.
>
> **Existing sample selection methods**: [1], one of the representative works, uses gradients to maintain the gradient diversity within the memory. [2] further devises a gradient-based scoring function that also considers adaptation to new tasks. [3] chooses memory samples that are most interfered with by the current task. [4] uses an information-theoretic metric within a Bayesian framework to balance information gain and learnability based on the estimated uncertainty. [5] samples memory per class using prototype vectors defined in the feature space.
>
> In contrast to these existing approaches, our method directly extracts memory without requiring labels and therefore leads to a more compact representation of past data. We will include a discussion in the main text to clarify these points and add citations to relevant works.
>
> References:
>
> [1] Gradient-based Sample Selection for Online Continual Learning – NeurIPS 2019
>
> [2] Online Coreset Selection for Rehearsal-based Continual Learning – ICLR 2022
>
> [3] Online Continual Learning with Maximally Interfered Retrieval – NeurIPS 2019
>
> [4] Information-theoretic Online Memory Selection for Continual Learning – ICLR 2022
>
> [5] iCaRL: Incremental Classifier and Representation Learning – CVPR 2017
>
> **Suggestion**: we have added a comparison to the method of [1] Aljundi (2019). Please refer to our response to [Question 3 (Q3) from Reviewer CdwX](https://openreview.net/forum?id=XLa5Puhqzg&noteId=z9EjRAahrW).

---

> > ### Comment · Reviewer_zWmk · 2025-08-03
> >
> > Thank you for following my suggestions and running additional experiments. I will increase my score from 4 to 5.
> >
> > **Some more comments:**
> >
> > - Please add additional discussion around the numerical stability and cost of PPCA EM and (truncated) SVD.
> > - I think it is worth highlighting that the choice of method to compute the compact memory (PPCA EM here) is orthogonal to the idea itself and that better suited (faster, more stable) methods might exist; this could be mentioned as potential future work.
> > - Could you share the actual wall-clock time overhead compared to experience replay for the new ImageNet-1k experiment and also include it in the paper?
> > - You might want to consider looking at and citing [1].
> >
> > [1] Ding et al. (2024). [Understanding Forgetting in Continual Learning with Linear Regression](https://arxiv.org/abs/2405.17583)

---

> ### Author Response · Authors · 2025-08-04
>
> Thank you for your additional comments and for updating your evaluation of our work. If you intended to change the score but did not, we kindly ask that you update it. We will incorporate your additional comments into the revised manuscript, including wall-clock time of the ImageNet-1k experiment.

---

### Official Review · Reviewer_ebAG · 2025-07-02

**Clarity:** 3
**Significance:** 3
**Originality:** 3
**Rating:** 4
**Confidence:** 3

**Summary:**

This paper presents a method to construct compact memory representations for continual learning with logistic regression, based on the K-prior framework. The key idea is to perform gradient reconstruction via Hessian matching, utilizing probabilistic PCA (PPCA) to iteratively extract compact memory vectors and their associated weights. The method generalizes from linear to logistic and multi-label regression and is shown to work effectively when using pretrained ViT feature extractors on datasets like CIFAR and TinyImageNet. The approach achieves significant memory efficiency while maintaining or exceeding baseline performance, especially under small memory constraints.

**Questions:**

Questions:
1. Why does SVD outperform EM at large data sizes (e.g., in Figure 2(a) at 320 samples)? A brief comment on this would help clarify the method’s relative strengths.

Suggestions:
1. The summation in Equation (1) uses index j, but the summand still depends on t, not j. This seems inconsistent and could benefit from correction.
2. On Line 135, … and 𝑓_𝑘^𝜃, defined similarly using 𝜃_𝑡, which should be: “ 𝑓_𝑘^(𝜃_𝑡), defined similarly using 𝜃_𝑡 ”
It would be helpful to cite the sources of the datasets used, such as USPS or Four-moon, for completeness and reproducibility.

**Ethical Concerns:**

["NO or VERY MINOR ethics concerns only"]

**Final Justification:**

Thanks for the rebuttal. My questions have been mostly addressed well. Overall, I believe the paper is technically sound. I maintain my original rating.

**Limitations:**

Yes.

**Paper Formatting Concerns:**

None.

**Quality:**

3

**Strengths And Weaknesses:**

Strengths:

The paper provides a principled and scalable solution for learning compact memory in continual learning, grounded in gradient/Hessian matching and the K-prior framework.

The use of Hessian matching and connection to the optimal gradient reconstruction theory is well-motivated and extended to logistic regression and multi-label classification.

The experimental results show that the proposed method is more effective than the previous approach, k-prior, as well as the random selection replay method, on CIFAR-10, CIFAR-100, and Tiny ImageNet-200.

Weaknesses:

The method currently applies only to last-layer adaptation using frozen feature extractors.

---

> ### Author Rebuttal · Authors · 2025-07-30
>
> We thank the reviewer for their encouraging comments.
>
> > Q1. Why does SVD outperform EM at large data sizes (e.g., in Figure 2(a) at 320 samples)? A brief comment on this would help clarify the method’s relative strengths.
>
> Thanks for the question. The difference in performance is mostly due to numerical issues (see [our response of Question 1 (Q1) to Reviewer zWmk](https://openreview.net/forum?id=XLa5Puhqzg&noteId=29gUNDP2KA)). For larger memory, SVD can get close to near perfect reconstruction and have less numerical issues.
>
> Thanks also for pointing out all the mistakes. We will fix them in the next version of our paper.

---

> > ### Comment · Reviewer_ebAG · 2025-08-06
> > **Reply to the rebuttal of authors**
> >
> > Thanks for the rebuttal. My questions have been mostly addressed. Overall, I believe the paper is technically sound. I would like to maintain my original rating.

---

### Official Review · Reviewer_CdwX · 2025-07-03

**Clarity:** 3
**Significance:** 3
**Originality:** 3
**Rating:** 5
**Confidence:** 4

**Summary:**

This paper proposes an algorithm on utilizing a compact memory for continual linear and logistic regression, which can be used in the last layer of neural networks.
The design of the compact memory is based on the theoretical properties of continual linear and logistic regression, and the efficiency of the algorithm is improved by the PPCA method.
The algorithm is evaluated on datasets including split MNIST, CIFAR and TinyImageNet.

**Questions:**

Q1: What is the computational complexity of each step of the algorithm of PPCA_EM_for_logistic? What is the complexity improvement compared to using PCA? Could you please use some specific examples (e.g., for the last layer of NN, there is a 2048 -> 10 classification) to illustrate how significant the complexity improvement is?

Q2: In experiments, I see that you compare with K-prior and standard replay with random sample selection. Could you please also compare with other sample selection methods, like [Aljundi, 2019]?

Q3: Is there any time analysis or ablation study on the time cost and performance comparison of your PPCA-based method and the one with PCA?

Q4: The algorithm is tested on datasets including TinyImageNet, with a similar number of images compared to MNIST and CIFAR. What is the reason for setting the experiment scale to this level? Is it possible to run it on the standard ImageNet?

[1] Aljundi, R., Lin, M., Goujaud, B., & Bengio, Y. (2019). Gradient based sample selection for online continual learning. Advances in neural information processing systems, 32.

**Ethical Concerns:**

["NO or VERY MINOR ethics concerns only"]

**Final Justification:**

Thank the author for their rebuttal. My concerns are addressed. I will increase my score from 4 to 5 and increase my confidence from 3 to 4.

I agree with most of Reviewer zWmk's suggestions. In particular, you might want to cite some theoretical CL work to justify the Hessian matching, and I leave some suggested papers:

[1] Evron et al. How catastrophic can catastrophic forgetting be in linear regression? COLT'22

[2] Li et al.  Fixed Design Analysis of Regularization-Based Continual Learning. CoLLAs'23

[3] Evron et al. Continual Learning in Linear Classification on Separable Data. ICML'23

[4] Li et al. Memory-statistics tradeoff in continual learning with structural regularization. arXiv:2504.04039.

**Limitations:**

N/A on societal impact; for technical see above.

**Quality:**

3

**Strengths And Weaknesses:**

I think this looks like a good paper. I have some questions on the algorithm and the experiments in the following Question part; if they are properly clarified I am happy to set the score to accept.

The proposed algorithm is quite simple in theory yet effective in practice, as illustrated in Sections 3 and 4. The writing is clear. The proposed algorithm is validated by comparing it with several prominent replay-based CL algorithms on several widely-used CL datasets.

I would like to seek some clarification on the computational complexity and the experimental setting. The authors claim that the computation via PPCA is faster than the PCA, but it is not very clear 1) what is the theoretical computational complexity and 2) what is the time cost in running the experiment.

---

> ### Author Rebuttal · Authors · 2025-07-30
>
> >Q1: What is the computational complexity of each step of the algorithm of PPCA_EM_for_logistic? What is the complexity improvement compared to using PCA? Could you please use some specific examples (e.g., for the last layer of NN, there is a 2048 -> 10 classification) to illustrate how significant the complexity improvement is?
>
> PCA is based on SVD which is costly compared to the EM iterations (both aim to solve Eq. 7). In our problem, SVD is applied to the concatenated matrix $[ \Phi_{t+1} , U_t ] \in R^\{P \times (T+K)\} $ with $P$ being the number of parameters, $T$ being the size of new data, and $K$ being the size of memory $U_t$; for example, we use $T=12000$, $P=784$, and memory size $K\in \\{10,20,30,40\\}$ (10 memory per task) in our experiment of Figure 2.
>
> The computational complexity of SVD is $\mathcal{O}( (T+K) P^{2} )$ for full SVD and $\mathcal{O}( (T+K) P K )$ for top-$K$ SVD.
>
> The computational cost of the EM algorithm is $\mathcal{O}(K^{3} )$ per iteration and is dominated by inverse of $K$-size square matrices $S$ and $(S^{-1}+ MM^{\top})$; see Algorithm 1. Since $K\ll T$ and $P$ both, this is faster if the number of iterations is reasonable.
>
> We note that the main reason to use EM is not always the speed but mainly its numerical stability compared to the version of SVD we used. It is possible to use other variants of SVD that can handle the numerical errors piling up as the new tasks are observed.
>
>
> ---
> > Q2: Is there any time analysis or ablation study on the time cost and performance comparison of your PPCA-based method and the one with PCA? Is experiment necessary?
>
> The performance of SVD was much worse (see Fig 2) which is why we did not conduct the time-analysis but we are happy to add it in the final version of the paper. EM was generally faster but not always, but it was definitely more numerically stable which is the reason why it worked better.
>
>
> ---
> > Q3: In experiments, I see that you compare with K-prior and standard replay with random sample selection. Could you please also compare with other sample selection methods, like [Aljundi, 2019]?
>
> Thanks for the suggestion. Below we report some comparison to Aljundi (2019)‘s method called Gradient-based Sample Selection (GSS) and find it to be worse than our method. We adapt the implementation provided by the authors into our setup of Figure 5. We consider several memory sizes from 1 to 20% and compared it to Experience Replay using GSS (ER-GSS) and Random sample selection (ER-RN). Additionally, we compare the ER using  class-wise random sample selection (ER-RN-C) that balances the label within the memory buffer.
>
> On CIFAR-10, the performance of GSS is slightly worse than that of random selection, but on CIFAR-100, it performs significantly worse, especially when the memory size is small (1% or 2%).
>
> | Dataset     |              |  1 %         | 2 %         | 5 %          | 10%        | 20%        |
> |-------------   |-------------|---------------|---------------|---------------|---------------|---------------|
> |                  | **Ours**     | **89.8 ± 0.2**    | **91.6 ± 0.2**    | **93.2 ± 0.1**    | **93.7 ± 0.1**    | **93.5 ± 0.1**    |
> | CIFAR-10  |**ER-RN-C**   | 87.9 ± 0.3    | 89.5 ± 0.2    | 91.2 ± 0.2    | 92.0 ± 0.3    | 92.3 ± 0.1    |
> |                  |**ER-RN**    | 87.9 ± 0.6    | 89.8 ± 0.5    | 91.4 ± 0.2    | 91.9 ± 0.2    | 92.3 ± 0.2    |
> |                  |**ER-GSS**    | 86.9 ± 0.4    | 88.8 ± 0.4    | 91.0 ± 0.2    | 91.8 ± 0.3    | 92.0 ± 0.2    |
>
>
> | Dataset     |              |  1 %         | 2 %         | 5 %          | 10%        | 20%        |
> |---------------|-------------|---------------|---------------|---------------|---------------|---------------|
> |                     | **Ours**      | **78.0 ± 0.4**    | **79.3 ± 0.2**    | **81.1 ± 0.2**    | **81.9 ± 0.1**    | **81.8 ± 0.2**    |
> | CIFAR-100  | **ER-RN-C**   | 56.6 ± 2.3    | 66.2 ± 0.6    | 72.9 ± 0.3    | 78.2 ± 0.2    | 81.8 ± 0.2    |
> |                     | **ER-RN**      | 51.3 ± 1.5    | 64.6 ± 1.3    | 72.9 ± 0.8    | 78.3 ± 0.2    | 81.5 ± 0.2    |
> |                     | **ER-GSS**      | 33.7 ± 2.5    | 54.8 ± 1.0    | 66.7 ± 1.0    | 73.3 ± 0.4    | 78.3 ± 0.2    |
>
> We hypothesize that this is because GSS fails to preserve label balance within the memory buffer because GSS  is known to select gradient-diverse samples. This issue is likely exacerbated when each task contains a large number of classes (e.g., 2 classes per task for CIFAR-10 vs. 10 classes per task for CIFAR-100). Indeed, the comparison between ER-RN-C and ER-RN on CIFAR-100 shows that the performance of ER-RN, which also does not consider label balance, was worse than that of ER-RN-C with 1% and 2% memory. This result supports our hypothesis on GSS’ performance degradation.
>
>
> ---
> > Q4: The algorithm is tested on datasets including TinyImageNet, with a similar number of images compared to MNIST and CIFAR. What is the reason for setting the experiment scale to this level? Is it possible to run it on the standard ImageNet?
>
> Thanks again for the suggestion. We used TinyImageNet-200 as the larger dataset, but now have also done experiments on ImageNet-1k where we find our method to again perform better. Following [1], we split the dataset into 10 tasks each with 100 consecutive classes and consider memory size of 0.5% (600 per task), 1% (1200 per task), and  2% (2400 per task) respectively. Below are the results:
>
> | #Memory | Replay        | K-Prior       | Ours             | Batch (Oracle) |
> |:-------:|:-------------:|:-------------:|:----------------:|:------:|
> | 0.5%    | 41.7 ± 0.2    | 36.3 ± 0.6    | **65.2 ± 0.2**    | 77.6   |
> | 1%      | 54.0 ± 0.4    | 49.4 ± 0.1    | **70.4 ± 0.0**    | 77.6   |
> | 2%      | 62.3 ± 0.1    | 59.9 ± 0.5    | **73.9 ± 0.2**    | 77.6   |
>
> We hope that our explanation and new experiments address your concerns. We are happy to further discuss these with you.
>
> [1] Improving Continual Learning by Accurate Gradient Reconstructions of the Past - TMLR 23

---

> ### Comment · Reviewer_CdwX · 2025-08-06
>
> Thank the author for their rebuttal. My concerns are addressed. I will increase my score from 4 to 5 and increase my confidence from 3 to 4.
>
> I agree with most of Reviewer zWmk's suggestions. In particular, you might want to cite some theoretical CL work to justify the Hessian matching, and I leave some suggested papers:
>
> [1] Evron et al. How catastrophic can catastrophic forgetting be in linear regression? COLT'22
>
> [2] Li et al.  Fixed Design Analysis of Regularization-Based Continual Learning. CoLLAs'23
>
> [3] Evron et al. Continual Learning in Linear Classification on Separable Data. ICML'23
>
> [4] Li et al. Memory-statistics tradeoff in continual learning with structural regularization. arXiv:2504.04039.

---

### Note · Authors · 2025-08-13

We thank all reviewers again for their helpful feedback and positive evaluations.
We are excited about our compact memory representation for continual learning, because it presents a way to learn memory representations for recovering gradients of past tasks and demonstrates significant memory efficiency improvements across various tasks, including ImageNet-1K. We therefore believe our work will be of high interest to the life-long continual learning research community at NeurIPS. Moreover, we believe that all questions have been addressed during the rebuttal period, and we will incorporate the discussions from the rebuttal into the revised manuscript. We sincerely hope our work is accepted at NeurIPS.
Thank all reviewers again for their thoughtful comments.

---

### Decision · Program_Chairs · 2025-09-17

**Decision:**

Accept (poster)

**Comment:**

[Overview]
This paper addresses the critical problem of compact memory for continual learning, specifically within the context of logistic regression and the K-prior regularization framework. It introduces a novel approach that formulates memory learning as a Hessian matching problem and solves it using a Probabilistic Principal Component Analysis (PPCA)-based Expectation Maximization (EM) algorithm. The reviews acknowledged the paper's interesting problem setting and the empirical results, with authors providing clarifications and commitments in the rebuttal that satisfactorily addressed key concerns.

[Strengths]
The paper presents a principled and novel methodology for generating compact memory representations by formulating the problem as Hessian matching and solving it via a PPCA-EM algorithm. This offers a theoretically grounded alternative to heuristic memory selection strategies.
Empirical evaluations demonstrate significant improvements in memory efficiency and performance over existing K-prior and experience replay baselines, especially in challenging multi-label classification tasks using pre-trained deep feature extractors (e.g., Split-CIFAR, Split-TinyImageNet), showcasing the method's practical utility.

[Weaknesses]
While the combination of techniques is novel, the conceptual justification for why Hessian matching is the optimal strategy for compact memory could be elaborated further to provide deeper theoretical insights.
The initial submission had some clarity issues in presentation and lacked complete experimental details.

[Recommendation]
The proposed Hessian matching approach for compact memory is a valuable contribution to continual learning, demonstrating strong empirical performance, particularly when integrated with modern pre-trained feature extractors. Although some clarity issues were noted, the authors' rebuttal adequately addressed the major concerns, making the paper a useful contribution to the field. Therefore, we recommend acceptance.